# Molecular embroidering of graphene

Tao Wei [1], Malte Kohring[2], Heiko B. Weber [2], Frank Hauke[1] & Andreas Hirsch [1✉]

Structured covalent two-dimensional patterning of graphene with different chemical functionalities constitutes a major challenge in nanotechnology. At the same time, it opens enormous opportunities towards tailoring of physical and chemical properties with limitless combinations of spatially defined surface functionalities. However, such highly integrated carbon-based architectures (graphene embroidery) are so far elusive. Here, we report a practical realization of molecular graphene embroidery by generating regular multiply functionalized patterns consisting of concentric regions of covalent addend binding. These spatially resolved hetero-architectures are generated by repetitive electron-beam lithography/reduction/covalent-binding sequences starting with polymethyl methacrylate covered graphene deposited on a Si/SiO$_2$ substrate. The corresponding functionalization zones carry bromobenzene-, deutero-, and chloro-addends. We employ statistical Raman spectroscopy together with scanning electron microscopy/energy dispersive X-ray spectroscopy for an unambiguous characterization. The exquisitely ordered nanoarchitectures of these covalently multi-patterned graphene sheets are clearly visualized.

[1] Department of Chemistry and Pharmacy & Joint Institute of Advance Materials and Processes (ZMP), Friedrich-Alexander University of Erlangen-Nuremberg (FAU), Nikolaus-Fiebiger-Strasse 10, 91058 Erlangen, Germany. [2] Department of Applied Physics & Institute of Condensed Matter Physics, Friedrich-Alexander University of Erlangen-Nuremberg (FAU), Staudtstrasse 7/Bau A3, 91058 Erlangen, Germany. ✉email: andreas.hirsch@fau.de

Multiply functionalized and hierarchically patterned graphene sheets with a spatially defined two-dimensional (2D)-architecture are still elusive. The realization of this challenge constitutes the next higher level of graphene chemistry. Such tailored nanosurfaces are highly attractive for high-performance applications, for example, in the fields of optoelectronics and sensors[1–4]. Key to this endeavor is providing facile and flexible synthetic approaches for 2D-patterning of graphene. So far, shape control over graphene nanostructures has been targeted either by top-down etching[5–10] or by bottom-up synthesis of nanoribbons using low-molecular precursors on surfaces[11–15]. Covalent graphene chemistry, in principle, provides another opportunity by addend binding to defined lattice regions. This approach also has the advantage that, next to the spatial structuring of graphene, chemical surface functionalities can be introduced which are located in the proximity of the conductive regions of intact graphene. To date, a few first examples of covalently 2D-patterned graphene have been prepared by a combination of wet-chemical and classical patterning techniques[16–25]. However, all covalent patterning protocols established so far are limited to one-step chemical patterning of graphene only. For the realization of more complex 2D-architectures involving the regular arrangement of addend regions with different surface functionalities, a detailed understanding of covalent reaction mechanisms and concepts such as surface activations are required. In the macroscopic world, such a 2D-engineering approach finds an analogy in embroidery. This is a long-existing decoration technique, in which a needle is used as a tool for applying patterns onto fabric and other materials. Following a similar manner, once the chemical needle (as referred to graphene chemistry) was manipulated, the analogous chemical embroidery will provide access to structured graphene nanoarchitectures with a multifunctional array of addend zones next to patterns of intact graphene. Owing to its versatility and chemical tailorability, lithography-assisted chemical functionalization[17,20] is assumed to be one important component for the realization of graphene embroidery. Very recently, we demonstrated a one-step patterned graphene functionalization protocol with diazonium salts involving lithography as a pattern defining tool[24]. We were able to fabricate well-defined nanostructures down to the submicrometer scale. Another pillar for this success was the realization of comparatively high degrees of functionalization in the addend-binding zones by using reductively activated graphene and allowing for substrate-mediated antaratopic additions.

Here, we realize graphene embroidery by generating regular multiply functionalized patterns, consisting of concentric regions with covalent addend binding. These spatially resolved 2D-hetero-architectures are generated by repetitive electron-beam lithography (EBL)/reduction/covalent-binding sequences starting with polymethyl methacrylate (PMMA)-covered graphene, deposited on a Si/SiO$_2$ substrate. We employ statistical Raman spectroscopy (SRS) together with scanning electron microscopy/energy dispersive X-ray spectroscopy (SEM-EDS) for an unambiguous characterization which clearly shows that covalently functionalized, multi-patterned graphene exhibits an exquisitely ordered nanostructure.

## Results

**Reaction sequence for the sequential pattern functionalization.** As a prototype for the sequential multiple functionalization of graphene we developed the generation of three concentric zones carrying bromobenzene-, deutero-, and chloro-addends (Fig. 1). These three functionalities were chosen in present study because: (I) The physical properties of the covalently grafted addends should be complementary (electron donating, electron withdrawing, neutral). (II) The introduction of addends, which could subsequently be exchanged by subsequent chemical transformations, can enhance the spectrum of potential applications, expanding the available addends that can be used for covalent patterning on graphene. (III) The introduced addends can be identified and characterized by the present set of analytical tools. The activation of the EBL-patterned graphene substrate towards addend binding was initiated by reduction with sodium/potassium (Na/K) alloy, an elegant strategy that we recently developed for the efficient covalent mono-patterning of graphene[24]. The commercially available graphene monolayer was deposited on a Si/SiO$_2$ wafer by a wet-transfer technique. Here, the PMMA-supported graphene, floating on top of a water surface, was fished onto the prepared Si/SiO$_2$ wafer and subsequently, the initial PMMA coating was removed by acetone and a fresh PMMA double-layer mask was applied by spin-coating. The usage of a reactive Si/SiO$_2$ substrate — reactivity provided by the presence of accessible surface atoms (O, etc.) — is a fundamental prerequisite as it allows for a strain-free antaratopic graphene addition scenario, which is in sharp contrast to supratopic addition reactions that would generate enormous strain in the graphene lattice[27–29]. By this technique, a comparatively high degree of functionalization can be established.

The circular patterns in the PMMA mask (array of $40 \times 40$ circles) for the subsequent three reductive activation/functionalization sequences were generated via EBL. After each individual functionalization step, the complete PMMA mask (which becomes partly damaged by the addition of the electron-trapping reagent benzonitrile) was completely removed by acetone vapor and a fresh PMMA double-layer mask has been applied via spin-coating to enable the next EBL patterning cycle (Supplementary section S2 and S3). In the first EBL patterning step, periodic dots with a hole diameter of 5 μm were generated (Fig. 1a) and the irradiated PMMA areas were cleaned by washing with an isopropanol-methyl/isobutyl ketone solution. In this way, uncovered graphene regions (zone I$_a$) are generated that are available for the first addend-binding process. The graphene in the PMMA–free zone I$_a$ was activated by a treatment with a Na/K alloy. Briefly, a drop of liquid Na/K alloy was dropped onto the graphene monolayer[24,27]. As a result, graphene was negatively charged, which increases the reactivity towards electrophiles. As soon as the Na/K alloy was removed by a gentle stream of argon, the efficient addend binding was accomplished by a reaction with 4-bromobenzene diazonium-tetrafluoroborate[30,31]. As a consequence, an array of the first dot-shaped addend zone I$_b$ was generated. The second concentric addend zones were generated following the analog concept. For this purpose, a second lithographic procedure was applied expanding the diameter of the concentric dots to 15 μm (zone II$_a$). After activation and exposure to deuteroxide, an efficient deuteration of the second patterning zone (zone II$_b$) was accomplished. This represents the first example for the generation of a deuterated monolayer graphene and remarkably, the achieved degrees of functionalization are comparable to that of deuterated bulk graphite and significantly exceed the numbers reported for deuterated multilayer graphene obtained by alternative functionalization approaches[32,33]. A third diameter expansion of the periodic dots to 20 μm (zone III$_a$) allows for the final concentric graphene patterning. Activation with Na/K and subsequent quenching with iodine monochloride (ICl) gave rise to a mild chlorination (no necessity of hazardous chlorine, Cl$_2$ gas[34–36]) generating the third addend zone (zone III$_b$). Strikingly, ICl was developed as a facile and mild chlorination source for graphene chlorination. The last step of the entire patterning procedure was the removal of the protective PMMA layer by a treatment with acetone. The final outcome was the first demonstration of a sequential multiple

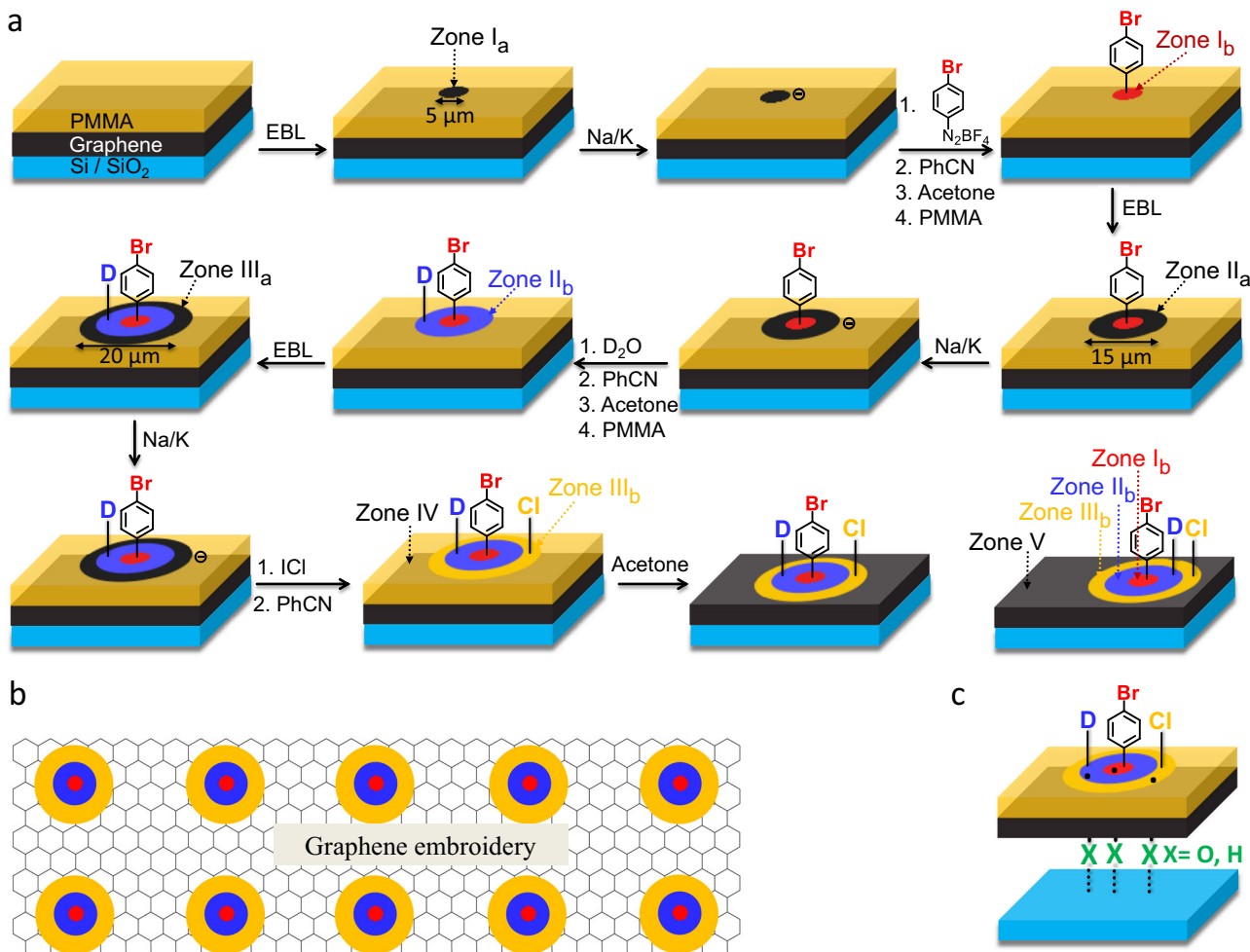

**Fig. 1 Reaction sequence towards multiply patterned graphene architectures. a** Schematic illustration of the reaction sequence for the multiple patterning functionalization of a graphene monolayer. PMMA: polymethyl methacrylate. EBL: electron-beam lithography; zone $I_a$, $II_a$, and $III_a$: the step-wise EBL exposed graphene areas (black circle with diameters of 5 μm, black concentric ring with diameters of 15 μm, and 20 μm) before functionalization; zone $I_b$, $II_b$, and $III_b$: reagent-accessible graphene areas after activation/functionalization. The first addend zone (zone $I_b$), showing covalently bound bromophenyl units, is indicated in red; the second addend zone (zone $II_b$), covalently bound deutero-atoms, is indicated in blue; the third addend zone (zone $III_b$), covalently bound chloro-atoms, is indicated in orange. Zone IV: PMMA-covered graphene and zone V: intact monolayer graphene after final PMMA removal. Each patterning process includes first spin-coating of a PMMA double-layer mask, and then EBL-based patterning of the graphene–PMMA assembly. The subsequent 2D-functionalization comprises: (1) addition of the trapping electrophile including 4-bromobenzene diazonium-tetrafluoroborate, deuteroxide, and iodine monochloride to graphene area reductively activated by sodium/potassium (Na/K) alloy; (2) quenching the reaction and the removal of residual charges by the addition of benzonitrile (PhCN);[26] (3) washing with acetone to remove unreacted chemicals as well as the old PMMA coverage. As this reagent leads to a partial degradation of the protective PMMA layer, the old PMMA coverage is removed by acetone. By this procedure, the initially created addend patterns (zone $I_b$, zone $II_b$) become covered by PMMA, but can be retrieved by the aid of a lithographically applied crosshair pattern, for details see supplementary section S2; and (4) re-spin-coating of a fresh PMMA layer. **b** Example image of the resulting graphene embroidery via molecular embroidering strategy. **c** During the entire reaction procedures, strain-free antaratopic additions provided by surface atoms (i.e., O) of the underlying Si/SiO2 substrate were enabled. Inert substrates would allow only for successive supratopic additions, which however are forbidden due to the enormous amount of strain energy that would be built up[27].

patterning of graphene with different addends bound to spatially defined addend zones. The concept can be considered as a molecular embroidery analogy to the traditional embroidery manufacturing, that is, employing spatially resolved graphene chemistry (needle) to embroider regularly patterned addend zones to graphene (fabric). With this pioneering concept various 2D-patterns beyond this concentric pattern can easily be accomplished by a tailor-made modification of the respective e-beam lithographic masks.

For our multiply patterned graphene, where the different functionalities including deuterium, chlorine, and bromobenzene were selectively bound to the graphene surface, the properties of

the constructed 2D-hetero-architecture can be tailored. Regular patterns of hydrogen/deuteration-covered regions result in a confinement potential for the carriers in the pristine graphene regions, thus leading to a bandgap opening and characteristic for a n-type semiconductor[3]. The bandgap depends on the amount of attached hydrogen/deuterium atoms. On the other hand chlorination causes p-doping[36]. By the attachment of differently substituted aromatic substituents ($\sigma$/$\pi$-donor or acceptor) a specific geometrically defined region of the graphene lattice can be addressed directly (here is aromatic acceptor) and fine-tuned with respect to its electrical properties. As a consequence, our constructed multiply patterned graphene with concentric

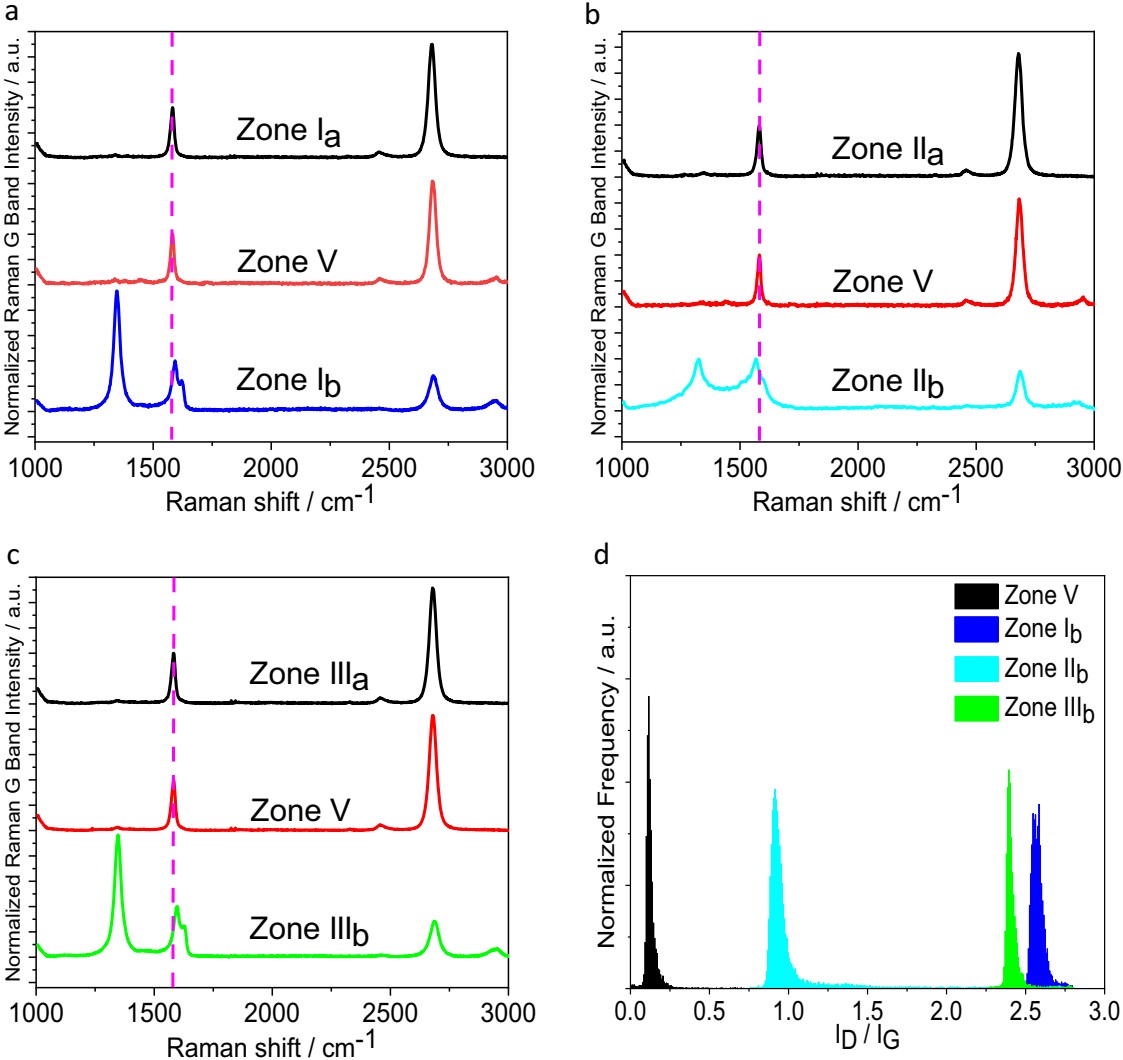

**Fig. 2 Statistical Raman analysis of a multiply patterned graphene monolayer. a** Raman spectra of zone $I_a$ (before arylation), zone V (after arylation and PMMA removal) and zone $I_b$ (after arylation). **b** Raman spectra of zone $II_a$ (before deuteration), zone V (after deuteration and PMMA removal) and zone $II_b$ (after deuteration). **c** Raman spectra of zone $III_a$ (before chlorination), zone V (after chlorination and PMMA removal) and zone $III_b$ (after chlorination). **d** Statistical Raman histograms of $I_D/I_G$ ratio extracted from Raman spectra of: (**a**) zone $I_b$ (blue), (**b**) zone $II_b$ (cyan), (**c**) zone $III_b$ (green), and (**d**) zone V (black). $I_D/I_G$: Raman D-band to G-band intensity ratio. Averaged spectra (~200 single point spectra) of the respective zones, $\lambda_{exc} = 532$ nm.

patterns, where each patterned graphene nanodomain has distinct semi-conducting behavior (n-type and p-type), bears promising potential to form graphene p–n junctions. Furthermore, the properties of these type of chemically written p–n junctions may further be fine-tuned by the control of the degree of functionalization in each addend zone (for details see section Thermal stability and reversibility). Semiconductor p–n junctions are elementary building blocks for many electronic devices such as transistors, solar cells, photodetectors, and integrated circuits and the potential applications of our constructed graphene p–n junctions in these fields can be reasonably expected. Besides, these constructed p–n junctions are always combined with the electric conductivity of the surround intact graphene regions (zone V), whose geometry can be tailored on demand upon the strategy that we introduce here. In view of this, the blueprint for promising all-graphene nanoscale electronic circuits is laid. Moreover, considering the fact that some aromatic systems are sensitive to light, the 2D-architectures we constructed can write phenyl-based substituents to distinct areas on the graphene flake, resulting in harvesting light of different wavelengths (based on the type of

attached chromophore) in different areas on the same electrical conducting substrate. Thus, it may also find applications in nanometer scale optoelectronics such as light sensor. Finally, our innovative approach of writing chemical information into spatially defined patterns may also become of great importance in a field of information storage based on distinct chemical molecules where more than only two states (on/off, which is the typical two-state electrical process of traditional information storage) of information can be stored.

**Statistical Raman analysis**. For the characterization of the concentrically patterned graphene sheets we applied SRS. The mean Raman spectra of the patterned zones are shown in Fig. 2. The zone $I_a$ (before functionalization) shows a pronounced G-band at 1582 cm$^{-1}$ indicative for a $sp^2$ carbon lattice along with a negligible defect-induced D-band at 1350 cm$^{-1}$. This is the characteristic feature of an intact monolayer graphene. The absence of a D-band demonstrates that during the EBL process no defects in the graphene lattice were generated. After the reductive arylation a very intense D-band emerged, demonstrating the efficient

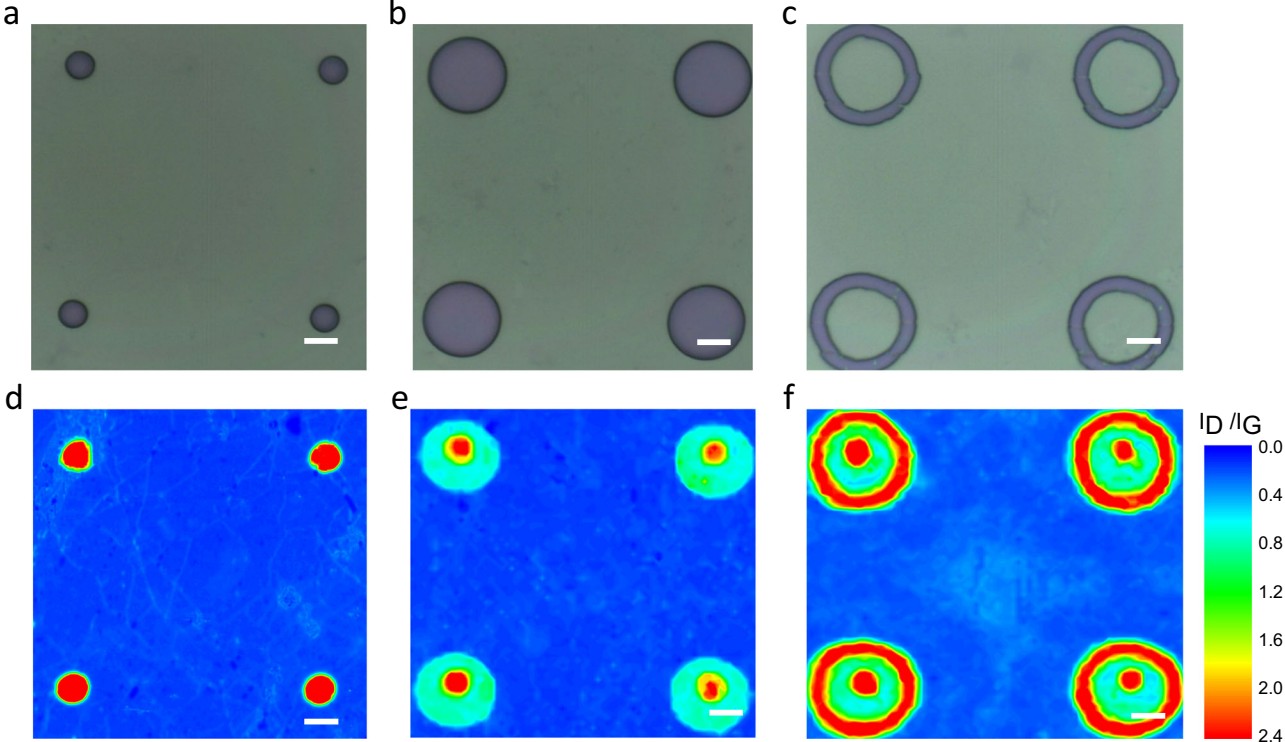

**Fig. 3 Optical images of pattern and large-scale Raman $I_D/I_G$ mapping. a, b, c** Optical image of PMMA (polymethyl methacrylate) patterned graphene (precursor masks) for each step. The purple areas are the exposed graphene regions and the light green areas are covered by PMMA. **d, e, f** The corresponding Raman $I_D/I_G$ mapping images after addend binding and removal of the PMMA layer. $I_D/I_G$: Raman D-band to G-band intensity ratio. The variations in color from blue to red in (**d, e, f**) represent the change in $I_D/I_G$ ratio from 0 to 2.4. The red circles (exposed regions, zone $I_b$) and blue background (covered regions, zone V) in (**d**) denote the $I_D/I_G$ ratios are 2.4 and 0, respectively. The red circles (zone $I_b$), turquoise concentric rings (zone $II_b$), and blue background (zone V) in (**e**) denote $I_D/I_G$ ratios are 2.4, 0.7, and 0, respectively. The red circles (zone $I_b$), turquoise concentric rings (zone $II_b$), red concentric rings (zone $III_b$), and blue background (zone V) denote $I_D/I_G$ ratios are 2.4, 0.7, 2.4, and 0, respectively. Scale bar = 5 μm.

conversion of $sp^2$ into $sp^3$ carbon atoms within the corresponding carbon lattice of zone $I_b$. The $I_D/I_G$ (Raman D-band to G-band intensity ratio) ratio raised from <0.1 to 2.6 after arylation (Fig. 2a, d). This high $I_D/I_G$ ratio indicates a high degree of functionalization, which is superior to previously reported covalent graphene patterning, emphasizing the advantage of our reductive protocol. The observed G-band at 1593 cm$^{-1}$ is upshifted relative to that measured before functionalization. This reflects the expected influence of the covalent binding of electron-withdrawing molecules to the remaining conjugated $\pi$-system in zone $I_b$[37,38]. Significantly, the unpatterned regions (zone V), which were covered by PMMA during the chemical treatment remains completely intact as no D-band was observed. Consequently, our graphene functionalization protocol selectively takes place only in the patterned region. To visualize this spatially resolved 2D-chemistry, large-scale Raman mapping was conducted and the corresponding pattern images can be clearly recognized through $I_D/I_G$ mapping (Fig. 3d). In addition, the rather uniform distribution of the $I_D/I_G$ ratio within the patterned region clearly speaks for a homogeneous addend binding.

For the zone $II_b$ a similar result is observed, that is, a high and very broad D-band with a $I_D/I_G$ ratio of 1. Again, the surrounding zone V remains unaffected and stays as defect-free monolayer graphene (Fig. 2b, d). It is noteworthy to mention that this very broad D-band is indicative of a rather high degree of functionalization with the corresponding $I_D/I_G$ ratio being located at the right side of the Cançado curve (high-functionalization-regime)[39]. In contrast, relative to the arylated graphene (zone $I_b$) deuteration causes n-doping of the remaining conjugated $\pi$-

system within zone $II_b$ as manifested by a downshift of the Raman G-band to 1567 cm$^{-1}$ (the G-band locates at 1582 cm$^{-1}$ before functionalization)[34,35]. The large-scale Raman $I_D/I_G$ mapping clearly reveals a turquoise pattern of the deuterated zone $II_b$ concentrically arranged around zone $I_b$ (Fig. 3e). The chlorinated zone $III_b$ was confirmed by Raman investigation as well as by the dramatically increased D-band with a $I_D/I_G$ ratio of 2.4 (Fig. 2c, d). Considering the electron-withdrawing effect of the Cl-atoms, the remaining conjugated $\pi$-system within zone $III_b$ is expected to show a p-type semi-conducting behavior. This was indeed corroborated by an upshift of the Raman G-band to 1592 cm$^{-1}$. The chlorination of zone $III_b$ is further reflected by a large-scale Raman $I_D/I_G$ mapping (Fig. 3f).

**Thermal stability and reversibility.** Temperature-dependent Raman spectroscopy was employed to investigate the thermal stability of these concentric graphene patterns. As shown in Fig. 4, the thermal stabilities of the zones $I_b$, $II_b$, and $III_b$ differ from each other significantly, which implies their individual chemical nature reflected in different strengths of the addend–graphene lattice bonds. For zone $I_b$, as temperature rises, the intensity of the D-band continuously decreases and eventually leads to a spectroscopic feature at 350 °C, which is the same as that for monolayer graphene (Fig. 4a). The thermally induced decline of the D-band is due to the restoration of $sp^2$ carbon lattice initiated by the cleavage of the C–C bonds between covalently linked aryl addends and graphene. The predominant defunctionalization process starts around 250 °C.

The thermal behavior of the deuterated zone $II_b$ is significantly different. In this case the intensity of the D-band keeps constant

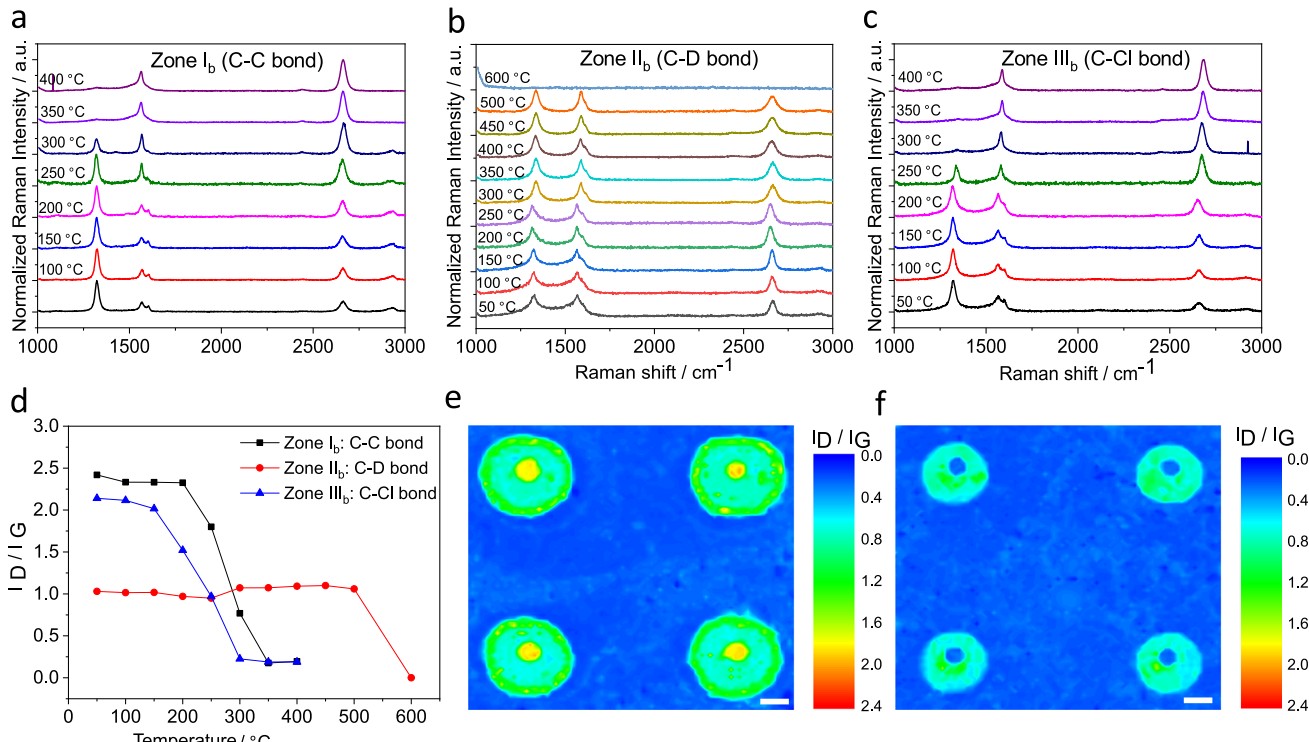

**Fig. 4 Temperature-dependent Raman analyses.** Temperature-dependent statistical Raman spectra of: **a** zone I$_b$, **b** zone II$_b$, and **c** zone III$_b$. **d** Mean Raman $I_D/I_G$ ratio extracted from the temperature-dependent Raman spectra of (**a**, **b**, **c**). Raman $I_D/I_G$ mapping after: **e** 250 °C annealing and **f** 400 °C annealing. The variations in color from blue to red in (**e**) and (**f**) represent the change in $I_D/I_G$ ratio from 0 to 2.4. The yellow circles, turquoise concentric rings, green concentric rings, and blue background in (**e**) denote $I_D/I_G$ ratios are 1.8, 0.7,1, and 0, respectively. The turquoise hollow rings and blue background in (**f**) denote $I_D/I_G$ ratios are 1 and 0, respectively. $I_D/I_G$: Raman D-band to G-band intensity ratio. Scale bar = 5 μm.

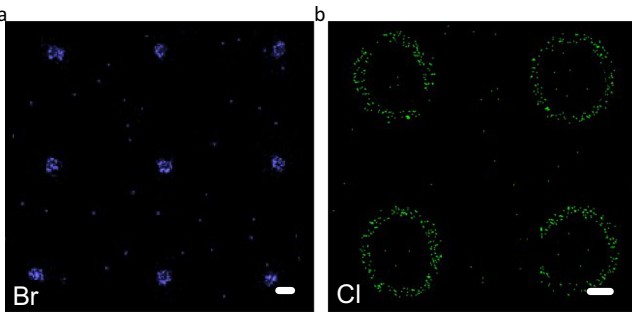

**Fig. 5 SEM-EDS analyses. a** EDS elemental mapping of bromine (Br, blue circles). **b** EDS elemental mapping of chlorine (Cl, green concentric rings). SEM-EDS: Scanning electron microscopy/energy dispersive X-ray spectroscopy. Scale bar = 5 μm.

up to 500 °C, indicating a very high thermal stability (Fig. 4b). Further increasing the temperature to 600 °C causes the complete loss of all Raman features including the G- and 2D-bands and the establishment of a flat line. This demonstrates the complete destruction of the graphene framework rather than just the cleavage of C–D bonds. Similar to zone I$_b$, the temperature-correlated D-band alteration was also found in zone III$_b$ and the D-band decreases continuously with increasing temperature (Fig. 4c). The decline of the D-band is a result of the C–Cl bond cleavage accompanied by $sp^3$ to $sp^2$ rehybridization. A closer look at the addend cleavage as a function of temperature nicely reflects the different thermal stabilities of zones I$_b$ and III$_b$. For the chlorinated zone III$_b$, the Raman features approaches that of pristine graphene at 300 °C instead of 350 °C, observed for the arylated zone I$_b$. Besides, the main addend cleavage sets in at

200 °C for zone III$_b$ and at 250 °C for zone I$_b$. The experimentally observed different thermal stabilities for zone I$_b$ (C–C bond cleavage), zone II$_b$ (C–D bond cleavage), and zone III$_b$ (C–Cl bond cleavage) correlate very well with the corresponding bond strengths (BE (C–Cl) = 328 kJ mol$^{-1}$, BE (C–C) = 332 kJ mol$^{-1}$, and BE (C–D) = 414 kJ mol$^{-1}$)[40–44].

As mentioned above, the reversible defunctionalization process mainly occurs in the temperature range between 200 and 350 °C for zone I$_b$ and III$_b$. Therefore, for instance, upon heating to 250 °C, we can tune the $I_D/I_G$ ratio of zone I$_b$ and zone III$_b$ to around 1.8 and 1, respectively. The large-scale Raman mapping clearly displayed the pattern image with the corresponding decreased $I_D/I_G$ ratios, respectively, encoded in yellow for zone I$_b$ and green for zone III$_b$ (Fig. 4e). The 400 °C heating treatment resulted in the complete defunctionalization of both zone I$_b$ and III$_b$ and leave only zone II$_b$ intact, which is confirmed by large-scale Raman mapping that the corresponding pattern image changes from the previous concentric circle to the hollow circle (Fig. 4f). Taking into account that the electronic character of the remaining conjugated π-system within a functionalized zone depends on the extent of functionalization such as chlorination, this reversible defunctionalization process enables the adjustment of the electronic properties (band gap engineering).

The defunctionalization behavior of this 2D-hetero-architectures also can be triggered by a re-reduction treatment with Na/K alloy (Supplementary section S6) and the different addend zones (I$_b$, II$_b$, and III$_b$) exhibit different defunctionalization processes (Supplementary Figs. 8–10). For zones I$_b$ and III$_b$, as the reduction time increases, the Raman D-band decreases continuously, which is indicative of an ongoing defunctionalization reaction. After 4 and 3 h of reductive treatment, respectively, the D-band of zone I$_b$ and zone III$_b$ have almost vanished,

suggesting a reversible defunctionalization processes. This is in clear line with similar studies carried out for carbon nanotubes[45], bulk graphene[46], and $C_{60}$ derivatives[47,48] and a clear indication that also in monolayer graphene systems, a post-functionalization reduction leads to a detachment of the previously introduced covalent functionalities. However, a completely different behavior was observed for deuterated graphene within zone $II_b$ (Supplementary Fig. 10). After 4 h and even up to 24 h of re-reduction treatment, the corresponding Raman D-band remains unchanged (Supplementary Fig. 11 and Supplementary Table 3), indicative for its high stability. These results are in line with the thermal treatment investigations.

**SEM-EDS analyses.** To track the chemical nature of this multiply patterned graphene, we carried out SEM-EDS (Fig. 5). Our patterned graphene containing the three addend zones $I_b$–$III_b$ contains the SEM-EDS detectable elements Br (zone $I_b$) and Cl (zone $III_b$). As expected, SEM-EDS investigation of the Br distribution in zone $I_b$ revealed periodic solid circular dot configurations correlating very well with the extension of zone $I_b$. When it comes to the element Cl, a distribution related to zone $III_b$, namely, a periodic ring configuration, is observed. The SEM-EDS results strongly correlate with the Raman data outlined above. In addition, the Br and Cl atoms are quite homogeneously distributed with the corresponding addend zones $I_b$ and $III_b$, demonstrating a homogeneous addend coverage. Moreover, no further element signals are found, which indicates the efficiency of all washing and work-up procedures and underlines the covalent nature of the patterned regions

## Discussion

In summary, we have realized the first prototype of multiply functionalized and hierarchically patterned graphene sheets with a spatially defined 2D-hetereo-architecture. The chemical embroidery was accomplished by repetitive EBL/reduction/ antaratopic-covalent-binding sequences starting with PMMA-covered graphene deposited on a Si/SiO₂ substrate. The corresponding functionalization zones carry bromobenzene-, deutero-, and chloro-addends. The successful covalent patterning was unequivocally demonstrated by SRS and SEM-EDS. In addition, the constructed concentric hetero-architectures were very clearly visualized by large-scale Raman $I_D/I_G$ mapping. Moreover, the different thermal stabilities for each addend zone were revealed by temperature-dependent Raman investigations and the reversibility of the covalent binding, as seen for the chloro- and bromobenzene-functionalities, provide a feasible means for property engineering (e.g., electronic structure). The construction principle of 2D-patterning that we have introduced here opens enormous opportunities towards tailoring the physical and chemical properties of graphene sheets with apparently limitless combinations of spatially defined surface functionalities. Such tailored nanosurfaces are highly attractive for high-performance applications, for example, in the fields of optoelectronics, sensors, and catalysis.

## Methods

**Covalent patterning and reductive functionalization of graphene.** The details on the multi-step covalent patterning and activation/functionalization of graphene and the involved chemicals are given in the Supplementary Information S2 and S3.

**Raman spectroscopy.** The Raman spectroscopic characterization was performed on a Horiba Jobin Yvon LabRAM Aramis. The spectrometer was calibrated by using crystalline graphite. All measurements were conducted using a laser (Olympus LMPlanFl50x, NA 0.50) with an excitation wavelength of 532 nm, with an acquisition time of 2 s. Spectral data were obtained through a motorized x-y table in a continuous line scan mode (SWIFT-module). The step sizes in the Raman mappings were kept in the range of 0.1–0.5 μm depending on the experiments. The corresponding data processing was performed using software of Lab

Spec 5. The temperature-dependent Raman measurements were performed in a Linkam stage THMS 600, equipped with a liquid nitrogen pump MS94 for temperature stabilization under a constant flow of nitrogen. Nitrogen gas needs to be filled for 20 min before starting the measurements to remove air. The measurements were carried out on Si/SiO₂ wafers with a heating rate of 10 K min⁻¹.

**SEM-EDS.** SEM-EDS was performed on a GeminiSEM 500 equipped with Oxford X-max 150. The working conditions were set at an operating at accelerating voltage of 5 kV, working distance of 7.3 mm, the elevation angle of detector is 35°, and the sample is vertical to the secondary-electron emission.

## Data availability
The authors declare that the data supporting the findings of this study are available within the article and its Supplementary Information files. All other relevant data supporting the findings of this study are available from the corresponding author on request.

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

## Acknowledgements
This work was financially supported by Deutsche Forschungsgemeinschaft (DFG, German Research Foundation) – Project number 182849149 – SFB 953.

## Author contributions
A.H. and T.W. conceived the research, designed the experiments and co-wrote the paper. A.H. and F.H. supervised the project as scientific group leader and principal investigator. T.W. synthesized the samples, performed Raman spectroscopy, SEM/EDS measurement and analyzed the data. M.K. and H.B.W. prepared the patterns on graphene. All the authors discussed the results and contributed to writing the manuscript.

## Funding

## Competing interests
The authors declare no competing interests.
