## [Peer Review File · Nature Communications]

Reviewer #1 (Remarks to the Author):

The manuscript reported the multiple step patterned functionalization of graphene by using PMMA as mask. Three different functionalization reactions have been demonstrated, and all of them start from Na/K treatment and followed by different functionalization reagents. The experiments are well-designed and the characterization results properly support the claimed patterned functionalization on graphene, or "graphene embroidery" as proposed by authors.

1. One major concern this reviewer raised is the novelty of the work. Previously, the patterned functionalization of graphene at the mask-defined area (some of the work used PMMA as mask) has been demonstrated, including the work by the author's group. The present work represents one step further to realized multiple step functionalization. However, the advancement is incremental and all the results are expected or at least not surprising.

2. The feature size in this work is at 5-10 micron scale. As can be seen from Figure 3, the patterned circles are obviously deviated from the designed concentric pattern. This is particularly surprising in view that EBL can routinely achieve resolution of tens of nm. Please comment on the alignment issue of the present method.

Overall, the work demonstrates the patterned functionalization of graphene. Scientifically, it is an incremental work and more suitable for a specialized journal. The demonstration of spatial resolution, fidelity and other metrics as well as specific application based on such a pattern technique would certainly increase the quality of the work.

Reviewer #2 (Remarks to the Author):

In this manuscript, titled "Molecular embroidering of graphene" by Wei, et al. the authors demonstrate a method to covalently modify graphene surfaces. The printing process involves depositing a PMMA resist onto the graphene basal plane and using e-beam lithography to create a pattern of areas where the graphene is exposed. UV-light treated PMMA-free graphene area was activated reductively by Na/K alloy which impart negative charge onto graphene surface. Subsequently 3 different electrophiles are applied step-wise to attach onto graphene monolayer. Bromobenzene diazonium-tetrafluoroborate, deuterioxide and iodine monochloride were chosen as reactants. Authors report differences in the thermal stability and reversibility of the 3 different adducts that were formed with the underlying graphene. Statistical Raman spectroscopy and scanning electron microscopy-energy dispersive X-ray spectroscopy (SEM-EDS) techniques were used for reaction monitoring and surface characterization. This is an interesting manuscript, the writing is clear, and the characterization is thorough. There are questions listed below that the authors should address before this manuscript should be considered for publication.

This work is the continuation of those studied previously in the PI's group. Their first attempt of reductive covalent functionalization of graphene was shown in 2011 (Nature Chem. 2011, 3, 279-286). The authors reported wet, bulk functionalization. Through effective reductive activation, covalent functionalization of the charged graphene is achieved by organic diazonium salts. In another work authors demonstrated reductive functionalization of graphene deposited on a surface. They interpreted differences in reactivity between monolayer and bilayer as a strain-free antaratopic bond formation between monolayer and silicon dioxide layer, which is suppressed in bilayer graphene (Angew. Chem. Int. Ed. 2016, 55, 14858). And very recently they have produced mono-patternes of graphene monolayer by derivatives of diazonium salts: 4-nitrobenzenediazonium-tetrafluoroborate and 4-bromobenzene diazonium-tetrafluoroborate (ref 24: Angew. Chem. Int. Ed. 2020, 59, 5602-5606). Performing all 3 reactions in a single pattern and comparing their relative stabilities is the novelty of this submitted manuscript.

Statistical Raman spectroscopy was exploited extensively for surface characterization. One unclear aspect in this manuscript is related to the procedural details: did the zone Ib was covered with PMMA before creating the zone IIa? Neither in the manuscript nor in the supporting information do not provide this information. Based on Figure 3 (A,B) we can assume that the zone Ib was not covered

with PMMA before creating the zone IIa. If this assumption is correct, does the formation of zone IIa affect the bonds in zone Ib? Are mixed monolayers formed?

The authors state "The experimentally observed different thermal stabilities for zone Ib (C-C bond cleavage), zone IIb (C-D bond cleavage) and zone IIIb (C-Cl bond cleavage) correlate very well with the corresponding bond strengths (BE (C-Cl) = 328 kJ/mol, BE (C-C) = 332 kJ/mol and BE (C-D) = 414 kJ/mol)". The authors should be able to extract kinetic data and possibly activation parameters, and these should be reported. This would require an estimate of the total functionalization of the graphene.

Generally, the novelty of this study is that patterns consisting of different components are realized.

The authors should define how such a pattern is useful. In other words, just because it can be made doesn't necessarily suggest that the resulting pattern is useful or is a groundbreaking advance.

Further the need for successive patterning and ex-situ introduction of reagents is cumbersome. Is this strategy amenable to any other patterning technique that would allow more pattern flexibility? The authors state "It can be expected that with this pioneering concept any type of complex 2D-pattern on graphene can be realized" but this is not at all obvious. Rather, they show chemistry, but there is no strategy presented for doing exactly what they state can be done, and the challenge is not trivial.

Using diazonium salts in covalently functionalization of carbon monolayer was used in other authors' work also (ELECTROPHORESIS, 2017, 38, 1669-1677; Carbon, 2017, 125, 49-55). The authors should include these references in any revisions.

In figure 1, some of the visuals are excessive and make the figure actually harder to understand.

Specifically, there is no need for the blue separated from the orange (see graphic in top right of figure). In similar work, this is not included, and the reason that it should be removed is that it just adds confusion without any clarification.

On page 3, the authors use the term "pronounced dueteration". It is not clear what distinction the authors are trying to make from previous dueteration.

Figure 4, and any images of patterns in manuscript or SI should have scale bars. This way the readers know the scale of the features.

Conclusions: prototype is one word

Reviewer #3 (Remarks to the Author):

This manuscript reports a sequential methodology for the concentric triple functionalization of graphene patterned surfaces. The controlled one-step chemical patterning of graphene has already been achieved by several groups, including a previous communication by the authors of the current work related with the reported research (ref. 24 of the manuscript). However, this work represents a breakthrough, allowing the step by step covalent modification of 2D patterned graphene by a combined electron-beam lithography/reductive protocol. The sequence of consecutive arylation, deuteration and chlorination reactions was monitored by statistical Raman spectroscopy, and the evaluation of the corresponding bond strengths investigated in thermal detaching experiments. Overall the research is solid and the presentation of results concise and clear with up to date references, only a few minor aspects might deserve further discussion/elaboration:

i) The authors state that a high degree of functionalization is achieved. Could this comment be quantified? Is the surface fully covered by the different functional groups? Differences in the reactivity of diazonium salts vs. D₂O vs. ICl must have been observed.

ii) In the legend of Figure 1 it is mentioned that "strain-free antaratopic additions provided by surface atoms (O, H) of the underlying SiO₂/Si substrate were enabled". This aspect, affecting reactivity, should be briefly mentioned in the text.

iii) After e-beam lithography graphene might be a reactive surface, at least for the arylation process. Control experiments before activation with Na/K should be carried out and discussed.

iv) In the cleaning process between subsequent functionalization processes benzonitrile is added to "remove the remaining negative charge on graphene". Does not remain attached to the surface?

Reviewer comments:

Reviewer #1 (Remarks to the Author):

The manuscript reported the multiple step patterned functionalization of graphene by using PMMA as mask. Three different functionalization reactions have been demonstrated, and all of them start from Na/K treatment and followed by different functionalization reagents. The experiments are well-designed and the characterization results properly support the claimed patterned functionalization on graphene, or “graphene embroidery” as proposed by authors.

1. One major concern this reviewer raised is the novelty of the work. Previously, the patterned functionalization of graphene at the mask-defined area (some of the work used PMMA as mask) has been demonstrated, including the work by the author's group. The present work represents one step further to realized multiple step functionalization. However, the advancement is incremental and all the results are expected or at least not surprising.

2. The feature size in this work is at 5-10 micron scale. As can be seen from Figure 3, the patterned circles are obviously deviated from the designed concentric pattern. This is particularly surprising in view that EBL can routinely achieve resolution of tens of nm. Please comment on the alignment issue of the present method.

Overall, the work demonstrates the patterned functionalization of graphene. Scientifically, it is an incremental work and more suitable for a specialized journal. The demonstration of spatial resolution, fidelity and other metrics as well as specific application based on such a pattern technique would certainly increase the quality of the work.

Reviewer #2 (Remarks to the Author):

In this manuscript, titled “Molecular embroidering of graphene” by Wei, et al. the authors demonstrate a method to covalently modify graphene surfaces. The printing process involves depositing a PMMA resist onto the graphene basal plane and using e-beam lithography to create a pattern of areas where the graphene is exposed. UV-light treated PMMA-free graphene area was activated reductively by Na/K alloy which impart negative charge onto graphene surface. Subsequently 3 different electrophiles are applied step-wise to attach onto graphene monolayer. Bromobenzene diazonium-tetrafluoroborate, deuterioxide and iodine monochloride were chosen as reactants. Authors report differences in the thermal stability and reversibility of the 3 different adducts that were formed with the underlying graphene.

Statistical Raman spectroscopy and scanning electron microscopy-energy dispersive X-ray spectroscopy (SEM-EDS) techniques were used for reaction monitoring and surface characterization.

This is an interesting manuscript, the writing is clear, and the characterization is thorough. There are questions listed below that the authors should address before this manuscript should be considered for publication.

This work is the continuation of those studied previously in the PI's group. Their first attempt of reductive covalent functionalization of graphene was shown in 2011 (Nature Chem. 2011, 3, 279-286). The authors reported wet, bulk functionalization. Through effective reductive activation, covalent functionalization of the charged graphene is achieved by organic diazonium salts. In another work authors demonstrated reductive functionalization of graphene deposited on a surface. They interpreted differences in reactivity between monolayer and bilayer as a strain-free antaratopic bond formation between monolayer and silicon dioxide layer, which is suppressed in bilayer graphene (Angew. Chem. Int. Ed. 2016, 55, 14858). And very recently they have produced mono-patternes of graphene monolayer by derivatives of diazonium salts: 4-nitrobenzenediazonium-tetrafluoroborate and 4-bromobenzene diazonium-tetrafluoroborate (ref 24: Angew. Chem. Int. Ed. 2020, 59, 5602-5606). Performing all 3

reactions in a single pattern and comparing their relative stabilities is the novelty of this submitted manuscript.

Statistical Raman spectroscopy was exploited extensively for surface characterization. One unclear aspect in this manuscript is related to the procedural details: did the zone Ib was covered with PMMA before creating the zone IIa? Neither in the manuscript nor in the supporting information do not provide this information. Based on Figure 3 (A,B) we can assume that the zone Ib was not covered with PMMA before creating the zone IIa. If this assumption is correct, does the formation of zone IIa affect the bonds in zone Ib? Are mixed monolayers formed?

The authors state "The experimentally observed different thermal stabilities for zone Ib (C-C bond cleavage), zone IIb (C-D bond cleavage) and zone IIIb (C-Cl bond cleavage) correlate very well with the corresponding bond strengths (BE (C-Cl) = 328 kJ/mol, BE (C-C) = 332 kJ/mol and BE (C-D) = 414 kJ/mol)". The authors should be able to extract kinetic data and possibly activation parameters, and these should be reported. This would require an estimate of the total functionalization of the graphene.

Generally, the novelty of this study is that patterns consisting of different components are realized. The authors should define how such a pattern is useful. In other words, just because it can be made doesn't necessarily suggest that the resulting pattern is useful or is a

groundbreaking advance. Further the need for successive patterning and ex-situ introduction of reagents is cumbersome. Is this strategy amenable to any other patterning technique that would allow more pattern flexibility? The authors state “It can be expected that with this pioneering concept any type of complex 2D-pattern on graphene can be realized” but this is not at all obvious. Rather, they show chemistry, but there is no strategy presented for doing exactly what they state can be done, and the challenge is not trivial.

Using diazonium salts in covalently functionalization of carbon monolayer was used in other authors' work also (ELECTROPHORESIS, 2017, 38, 1669-1677; Carbon, 2017, 125, 49-55). The authors should include these references in any revisions.

In figure 1, some of the visuals are excessive and make the figure actually harder to understand. Specifically, there is no need for the blue separated from the orange (see graphic in top right of figure). In similar work, this is not included, and the reason that it should be removed is that it just adds confusion without any clarification.

On page 3, the authors use the term “pronounced dueteration”. It is not clear what distinction the authors are trying to make from previous dueteration.

Figure 4, and any images of patterns in manuscript or SI should have scale bars. This way the readers know the scale of the features.

Conclusions: prototype is one word

Reviewer #3 (Remarks to the Author):

This manuscript reports a sequential methodology for the concentric triple functionalization of graphene patterned surfaces. The controlled one-step chemical patterning of graphene has already been achieved by several groups, including a previous communication by the authors of the current work related with the reported research (ref. 24 of the manuscript). However, this work represents a breakthrough, allowing the step by step covalent modification of 2D patterned graphene by a combined electron-beam lithography/reductive protocol. The sequence of consecutive arylation, deuteration and chlorination reactions was monitored by statistical Raman spectroscopy, and the evaluation of the corresponding bond strengths investigated in thermal detaching experiments. Overall the research is solid and the presentation of results concise and clear with up to date references, only a few minor aspects might deserve further discussion/elaboration:

i) The authors state that a high degree of functionalization is achieved. Could this comment be quantified? Is the surface fully covered by the different functional groups? Differences in the reactivity of diazonium salts vs. D₂O vs. ICl must have been observed.

ii) In the legend of Figure 1 it is mentioned that “strain-free antaratopic additions provided by surface atoms (O, H) of the underlying SiO₂/Si substrate were enabled”. This aspect, affecting reactivity, should be briefly mentioned in the text.

iii) After e-beam lithography graphene might be a reactive surface, at least for the arylation process. Control experiments before activation with Na/K should be carried out and discussed.

iv) In the cleaning process between subsequent functionalization processes benzonitrile is added to “remove the remaining negative charge on graphene”. Does not remain attached to the surface?

Point-By-Point Response:

Reviewer 1: “The manuscript reported the multiple step patterned functionalization of graphene by using PMMA as mask. Three different functionalization reactions have been demonstrated, and all of them start from Na/K treatment and followed by different functionalization reagents. The experiments are well-designed and the characterization results properly support the claimed patterned functionalization on graphene, or “graphene embroidery” as proposed by authors.”

- We thank the referee for her/his time to review our manuscript and her/his insightful comments for an improvement of our manuscript.

Comment 1: “One major concern this reviewer raised is the novelty of the work. Previously, the patterned functionalization of graphene at the mask-defined area (some of the work used PMMA as mask) has been demonstrated, including the work by the author’s group. The present work represents one step further to realized multiple step functionalization. However, the advancement is incremental and all the results are expected or at least not surprising.”

Answer: The reviewer is right up to the point that mask-assisted patterning of graphene has been studied already and several precedents have been reported in the literature (Z. Sun *et al.*, *Nat. Commun.* **2011**, *2*, 559-564; F. M. Koehler *et al.*, *Angew. Chem. Int. Ed.* **2009**, *48*, 224-227; J. Li *et al.*, *J. Am. Chem. Soc.* **2016**, *138*, 7448-7451). Nevertheless, the controlled covalent functionalization – and especially multi-functionalization – of graphene localized in specific, predefined areas remains an unsolved challenging task due to two major drawbacks of previous approaches.

One obvious shortcoming is the moderately low degree of functionalization inherent in the reported approaches and which now can be overcome by our recently established activation concept of substrate-deposited monolayer graphene (T. Wei *et al.*, *Angew. Chem. Int. Ed.* **2020**, *59*, 5602-5606 as also mentioned by reviewer). Another even more important drawback is that the addends currently available for patterning graphene are very limited (restricted to hydrogen-, aryl-, and cis-diene entities, (Z. Sun *et al.*, *Nat. Commun.* **2011**, *2*, 559-564; F. M. Koehler *et al.*, *Angew. Chem. Int. Ed.* **2009**, *48*, 224-227; J. Li *et al.*, *J. Am. Chem. Soc.* **2016**, *138*, 7448-7451). Therefore, it remains a fundamental task to explore and elaborate synthetic strategies to introduce a broader portfolio of addends for covalent addend

binding in patterned graphene architectures (such as *cis*-diene addend, J. Li *et al.*, *J. Am. Chem. Soc.* **2016**, *138*, 7448-7451) – in our present study we expanded that portfolio by the reductive addition of deuterium and the addend binding of chlorine atoms by the utilization of iodine monochloride (two addends which have been made available for the first time for graphene patterning).

More importantly, the precise assembly of different addends, in tailor-made patterns, on distinct areas of the graphene surface – generating up to now elusive multiply-patterned-graphene architectures – is undoubtedly of uttermost significance for a brought readership and for the technological implementation of these kind of materials in future applications. Beyond this point, our findings now open up the possibility to fine-tune the degree of functionalization of each individual addend zone by a post functionalization thermal annealing step. As we have shown, the C-addend bond strength correlates with the temperature required for the subsequent partial detachment of the functional entities – this possibility has not been described or addressed by preceding publications. Altogether, with our submitted work we present a successful and comprehensive approach for addressing all the challenges described above and in our opinion this work goes far beyond the investigations reported up to now in literature. This generation of up to present elusive 2D-hetero-architectures represents a breakthrough in graphene research and processing.

Comment 2: “The feature size in this work is at 5-10 micron scale. As can be seen from Figure 3, the patterned circles are obviously deviated from the designed concentric pattern. This is particularly surprising in view that EBL can routinely achieve resolution of tens of nm. Please comment on the alignment issue of the present method.”

Answer: The reviewer is right that the resolution of EBL can reach down to the nanometer scale. Actually, the slight deviation of the generated patterns (shown in Fig. 3) from a perfect concentric shape is based on the way how the predefined patterns on the PMMA mask were generated (it's not the problem of EBL resolution). These concentric patterns are generated by an iterative patterning procedure of suitable PMMA masks *via* EBL and each individual patterning/functionalization step of the process is accompanied by a complete removal of the PMMA layer and the spin-coating of a fresh PMMA double-layer to enable the next EBL patterning cycle. We would like to apologize that this important point was not explicitly addressed in our initial manuscript and we have now added this information to the revised version of the manuscript and the electronic supplementary information.

By the implementation of a fresh PMMA double-layer, the created addend patterns become covered by the freshly deposited PMMA film and thus the localization of the chemically

functionalized areas turns out to be tricky. In order to determine the exact positions of the tailor-made pattern regions and therefore of the covalently functionalized zones we initially applied a crosshair pattern (with additional letters for a better orientation) on the graphene by lithographic techniques (see Fig S1). Based on this set of distinct markers, the position for the subsequent EBL patterning spots can be located and the respective concentric shapes can be “written”. Unfortunately, the localization of the preceding patterning spots cannot be addressed in the high accuracy scale of the EBL resolution.

Moreover, the focus of our study was the demonstration/proof of concept of the embroidery technique. Less attention has been paid on position accuracy. In order to achieve this goal we patterned an array of 40 x 40 circles, for which a rather large writing field was the natural choice. As a consequence, the reposition accuracy is only in the micrometer scale. If special care is taken on the reposition accuracy, one would rather design and align smaller write fields (2 x 2 circles) for which we would expect indeed accuracies below 100 nm in x- and y-coordinates. By stitching techniques also larger areas can be accurately decorated, which was not the intention of this proof-of-principle manuscript. Based on these two reasons, our predesigned patterns of concentric rings deviate from perfect concentric patterns that could in principle be achieved by the EBL resolution. We thank the reviewer for bringing our attention to that basic point and we have therefore added the respective information in the experimental section of the electronic supplementary information.

Comment 3: “Overall, the work demonstrates the patterned functionalization of graphene. Scientifically, it is an incremental work and more suitable for a specialized journal. The demonstration of spatial resolution, fidelity and other metrics as well as specific application based on such a pattern technique would certainly increase the quality of the work.”

Answer: As we have stated above, our novel approach for the generation 2D-hetero-architecture based on graphene monolayers is of uttermost significance and is of broad interests for the scientific community.

In principle and as outlined by the reviewer, nanoscale resolutions (~ 30 nm for periodic structures) can be reached by electron beam lithography (C. Vieu, F. Carcenac, A. Pépin, Y. Chen, M. Mejias, A. Lebib, L. Manin-Ferlazzo, L. Couraud, H. Launois, “Electron beam lithography: resolution limits and applications”, *Applied Surface Science*, Volume 164, Issue 1, p. 111-117). Writing of such nanoscale patterns inherently raises fundamental questions with respect to the subsequent chemical functionalization steps. For instance, it is not obvious that the chemical reagents needed for the covalent activation/functionalization of the graphene lattice even can reach down to the surface in such narrow resist pits. It has also to

be expected that there are limitations due to wetting problems and/or capillary effects. Another issue which has to be solved, when one goes down to a narrower spatial resolution, is that the protective PMMA resist sufficiently sticks to the graphene surface. It has to be ensured that the reagents do not creep under the resist and therefore a tailor-made patterning on a nanoscale resolution regime becomes inaccessible. Altogether, going to narrower patterns with higher spatial resolution will be a fundamental technological challenge and represents in principle a fine-tuning of our general patterning/functionalization protocol, which goes beyond the scope of the present manuscript.

The other mentioned point by the referee - the technological application of this multiply patterned graphene architectures – is undoubtedly very interesting. Considering that covalent graphene patterning is still in its infancy, our fundamental exploration of the strategy for covalent patterning of graphene towards various 2D-hetero-architectures represents an important accomplishment. In our synthesized multiply-patterned-graphene, the different functionalities attached to the spatially restricted areas of the graphene lattice exhibit different physical properties (i.e. electro-withdrawing and electro-donating). Therefore, with our presented versatile tool for the generation of a chemically patterned graphene surface, the properties of the respective 2D-hetero-architectures can not only be tailored but also greatly enriched. We are convinced that our discovery has great potential in opto-electronics, molecular recognition as well as for magnetic-, catalytic- and sensor applications (as we have already stated in the manuscript, page 4). Investigations towards practical applications of this kind of novel 2D-hetero-architectures and their implementation in respective devices are currently under investigation in our laboratory and hopefully will be reported in due course.

Reviewer 2: In this manuscript, titled “Molecular embroidering of graphene” by Wei, et al. the authors demonstrate a method to covalently modify graphene surfaces. The printing process involves depositing a PMMA resist onto the graphene basal plane and using e-beam lithography to create a pattern of areas where the graphene is exposed. UV-light treated PMMA-free graphene area was activated reductively by Na/K alloy which impart negative charge onto graphene surface. Subsequently 3 different electrophiles are applied step-wise to attach onto graphene monolayer. Bromobenzene diazonium-tetrafluoroborate, deuteroxide and iodine monochloride were chosen as reactants. Authors report differences in the thermal stability and reversibility of the 3 different adducts that were formed with the underlying graphene. Statistical Raman spectroscopy and scanning electron microscopy-energy dispersive X-ray spectroscopy (SEM-EDS) techniques were used for reaction monitoring and surface characterization.

This is an interesting manuscript, the writing is clear, and the characterization is thorough. There are questions listed below that the authors should address before this manuscript should be considered for publication.

This work is the continuation of those studied previously in the PI's group. Their first attempt of reductive covalent functionalization of graphene was shown in 2011 (Nature Chem. 2011, 3, 279-286). The authors reported wet, bulk functionalization. Through effective reductive activation, covalent functionalization of the charged graphene is achieved by organic diazonium salts. In another work authors demonstrated reductive functionalization of graphene deposited on a surface. They interpreted differences in reactivity between monolayer and bilayer as a strain-free antaratomic bond formation between monolayer and silicon dioxide layer, which is suppressed in bilayer graphene (Angew. Chem. Int. Ed. 2016, 55, 14858). And very recently they have produced mono-patternes of graphene monolayer by derivatives of diazonium salts: 4-nitrobenzenediazonium-tetrafluoroborate and 4-bromobenzene diazonium-tetrafluoroborate (ref 24: Angew. Chem. Int. Ed. 2020, 59, 5602-5606). Performing all 3 reactions in a single pattern and comparing their relative stabilities is the novelty of this submitted manuscript.

- We are very thankful for this reviewer's very positive feedback and for her/his precious and thoughtful suggestions, which helped us to improve our manuscript. We thank the referee for the kind response on our manuscript in relation to the earlier work in the field of carbon allotrope functionalization.

Comment 1: “Statistical Raman spectroscopy was exploited extensively for surface characterization. One unclear aspect in this manuscript is related to the procedural details: did the zone Ib was covered with PMMA before creating the zone IIa? Neither in the manuscript nor in the supporting information do not provide this information. Based on Figure

3 (A,B) we can assume that the zone Ib was not covered with PMMA before creating the zone IIa. If this assumption is correct, does the formation of zone IIa affect the bonds in zone Ib? Are mixed monolayers formed?.”

Answer: The referee is right with his/her assumption that the patterned PMMA mask is removed after each covalent functionalization sequence and a fresh PMMA double-layer is installed prior to the next EBL-patterning cycle. This is due to the fact that the PMMA layer becomes partly damaged by the addition of the electron-trapping reagent benzonitrile.

We thank the referee for bringing our attention to this highly important point and we are sorry that we didn't outline the novel patterning/functionalization procedure in every single detail. This has now been corrected and we added substantial information to the main manuscript and to the electronic supplementary information to clarify each process step.

First, we have now substantially revised Figure 1 and the respective caption text. (see revised manuscript). Second, in the experimental section the given information has substantially been extended and the implementation of the PMMA mask and the removal has been discussed with respect to each patterning step.

Regarding the second part of the question: After the first activation/functionalization by 4-bromobenzenediazonium tetrafluoroborate (arylation), the PMMA mask was removed and a fresh PMMA double-layer was installed. The respective functionalization zone I_b has been localized by a set of crosshair markers (see also next comment) and in the second EBL patterning step, periodic dots with a hole diameter of 15 μm were “written” into the PMMA mask. Therefore, when creating the second patterns (zone II_a), the first pattern (zone I_b) was simultaneously exposed by etching away the PMMA (see Figure 3B). As it is demonstrated by the SEM-EDS analyses of the final multi-functionalized sample (Figure 5 in the main manuscript) this lithographic removal of the PMMA protective layer does not lead to a removal of the 4-bromobenzene addends introduced in the first reductive activation/functionalization step (zone I_b). This was also demonstrated by Raman results (see Figure R1) that the I_D/I_G of zone I_b (undergoing EBL treatment) remains unchanged. Moreover, the re-reduction of the addend zone I_b within the Na/K ally activation step for the functionalization of zone II_a does not lead to a pronounced retro-functionalization as demonstrated by the respective reference experiments presented and discussed in the electronic supplementary information.

A point which cannot be excluded is that an addition of deuterium atoms may also take place in addend zone I_b as this is fully accessible by all reagents – a point which has been taken into account in our third patterning/functionalization step, where only a ring type region has been exposed by EBL. Nevertheless, considering the advantage of our direct reductive

activation concept, providing a very high degree of functionalization, zone I_b has been highly functionalized – probably up to saturation, as an excess of the activation and crafting reagent has been used. As a consequence, the deuterium-atoms (during the second-step of reaction) should preferentially be bound in zone II_a rather than zone I_b. Therefore, the zone I_b is also not affected during the second-step of patterning deuteration of zone II_a to form zone II_b. This observation is further corroborated by the respective temperature-dependent investigations. Upon annealing to 400 °C, the zone I_b was completely defunctionalized, whereas zone II_b (deuterated) remained intact. If zone I_b was partially deuterated resulting from second-step of deuteration of zone I_a to form zone I_b, the zone I_b should not be completely defunctionalized.

Figure R1. The Raman spectra of zone I_b before and after EBL treatment.

Comment 2: “The authors state “The experimentally observed different thermal stabilities for zone Ib (C-C bond cleavage), zone IIb (C-D bond cleavage) and zone IIIb (C-Cl bond cleavage) correlate very well with the corresponding bond strengths (BE (C-Cl) = 328 kJ/mol, BE (C-C) = 332 kJ/mol and BE (C-D) = 414 kJ/mol)”. The authors should be able to extract kinetic data and possibly activation parameters, and these should be reported. This would require an estimate of the total functionalization of the graphene.”

Answer: We totally agree with the reviewer that both, thermodynamic and kinetic parameters, will affect the bond breaking between graphene and the respective addends in the course of the temperature dependent Raman experiments.

The respective parameters for arylated graphene (diazonium functionalization), deuterated graphene and chlorinated graphene have been deeply and systematically investigated in previous investigations (A. Paris *et al.*, *Adv. Funct. Mater.* **2013**, *23*, 1628-1635; A. Sinitskii *et al.*, *ACS Nano*, **2010**, *4*, 1949-1954; M. Ijäs *et al.*, *Phys. Rev. B*, **2012**, *85*, 035440-035450; X. Y. Liu *et al.*, *Physic B*, **2014**, *436*, 54-58; M. Pykal *et al.*, *Phys. Chem. Chem. Phys.*, **2016**, *18*,

6351-6372). Our experimentally observed defunctionalization behavior of the patterned 2D-hetero-architectures is very consistent with these results, which clearly supports our assumptions. In order to support our conclusions, we have now added those abovementioned citations (ref. 41 – ref. 45) in the revised version of the manuscript.

Comment 3: “Generally, the novelty of this study is that patterns consisting of different components are realized. The authors should define how such a pattern is useful. In other words, just because it can be made doesn’t necessarily suggest that the resulting pattern is useful or is a groundbreaking advance. Further the need for successive patterning and ex-situ introduction of reagents is cumbersome. Is this strategy amenable to any other patterning technique that would allow more pattern flexibility? The authors state “It can be expected that with this pioneering concept any type of complex 2D-pattern on graphene can be realized” but this is not at all obvious. Rather, they show chemistry, but there is no strategy presented for doing exactly what they state can be done, and the challenge is not trivial.”

Answer: To explore and implement practical applications of functional graphene materials, which is certainly the long-term goal, a suitable and robust chemical tool of patterned 2D-functionalization has to be elaborated and implemented first. The successful exploration of this prerequisite was the purpose of this study. It is worth emphasizing that the chemical patterning of graphene is still in its infancy. Fundamental investigations on exploring more addends of different chemical nature and in particular the successive and spatially resolved grafting to graphene thus constructing complex 2D-hetero-architectures are in any case very important. For our multiply-patterned-graphene, where the different functionalities (including two newly discovered addends) possessing different properties (electro-withdrawing and electro-donating) were selectively bound to the graphene surface, the properties of the constructed 2D-hetero-architecture can not only be tailored but also greatly enriched. Thus versatile potential applications in i.e. opto-electronics, molecular recognition, magnetic- and sensor devices can be expected (as already outline in our manuscript page 4). Nevertheless, investigation towards practical applications of such 2D-hetero-architectures and their device implementation are currently underway in our laboratory and beyond the scope of this manuscript.

Our proposed reaction sequence can in principle be combined with many other lithography techniques such as nanowire lithography, nanosphere lithography, and nanoimprinted lithography (in these methods the nanostructures can serve as mask), to achieve covalent patterning of graphene offering a very broad variety of different patterns.

We agree with the reviewer that it's a little bit risky to state that **any** types of complex 2D-pattern on graphene can be realized based on our introduced concept. Herein, on the one hand, we want to emphasize that in addition to the electrophilic reagents mentioned in this article and our previously reported examples (alkyl-, aryl-halides, or λ^3 -iodanes *etc.* F. Hof *et al.*, *Chem.-Eur. J.* **2014**, *20*, 16644-16651; K. Knirsch *et al.*, *Chem. Commun.* **2013**, *49*, 10811-10813; R. Schäfer *et al.*, *Angew. Chem., Int. Ed.* **2016**, *55*, 14858-14862; G. Abellán *et al.*, *J. Am. Chem. Soc.* **2017**, *139*, 5175-5182), our reductive activation strategy can in principle allow for being used in a wider scope of electrophiles to be added onto the graphene surface (the possibilities are very exciting). On the other hand, it's well accepted that arbitrary patterns can be made on graphene *via* EBL (Y. Q. Zheng *et al.*, *Adv. Mater. Technol.* **2017**, *2*, 1600237-1600249. A. Hirsch *et al.*, *ChemPlusChem*, 2020, doi.org/10.1002/cplu.202000419, online). Altogether, with our reaction sequence involving EBL, various (though not any) 2D-patterns on graphene can be certainly expected. Based on this consideration, we rephrased the original sentence into "*With this pioneering concept various 2D-patterns beyond this concentric pattern can easily be accomplished by a tailor-made modification of the respective e-beam lithographic masks.*" (see page 4, revised manuscript)

Comment 4: "Using diazonium salts in covalently functionalization of carbon monolayer was used in other authors' work also (ELECTROPHORESIS, 2017, 38, 1669-1677; Carbon, 2017, 125, 49-55). The authors should include these references in any revisions."

Answer: We thank the reviewer for this very good suggestion. These two references have been added and the references in manuscript have been renumbered accordingly.

Comment 5: "In figure 1, some of the visuals are excessive and make the figure actually harder to understand. Specifically, there is no need for the blue separated from the orange (see graphic in top right of figure). In similar work, this is not included, and the reason that it should be removed is that it just adds confusion without any clarification."

Answer: Following the reviewer's suggestion the binding topologies of the antaratopic addition mode were removed from Figure 1A and placed separately as Figure 1C. Furthermore, as stated above, Figure 1 and the respective caption text has substantially been improved by giving additional experimental information for the individual patterning/functionalization steps.

Comment 6: "On page 3, the authors use the term "pronounced deuteration". It is not clear what distinction the authors are trying to make from previous dueteration."

Answer: The reductive activation concept enables a high degree of deuteration of graphene which we have demonstrated with **bulk** graphene chemistry based on potassium intercalated graphite as starting material (R. Schäfer *et al.*, *J. Am. Chem. Soc.* **2016**, *138*, 1647-1652). In

the present study, we report for the first time on the reductive deuteration of **monolayer** graphene. This allowed us to achieve high degrees of functionalization, which is in sharp contrast to recently reported deuteration of multi-layer graphene without activation (A. R. Mazza *et al.*, *J. Vac. Sci. Technol. B*, **2019**, 37, 041804). Taking into account of reviewer's suggestion, we rephrased the original sentences into "*This represents the first example for the generation of deuterated monolayer graphene and remarkably, the achieved degrees of functionalization are comparable to that of deuterated bulk graphite and significantly exceed the numbers reported for deuterated multi-layer graphene obtained by alternative functionalization approaches.*^{33,34}" and the corresponding reference has been added to the revised version of the manuscript.

Comment 6: "Figure 4, and any images of patterns in manuscript or SI should have scale bars. This way the readers know the scale of the features."

Answer: We thank the reviewer for this suggestion and apologize for having overlooked this point in the first place - scale bars were accordingly added (see revised version of the manuscript).

Comment 7: "Conclusions: prototype is one word."

Answer: Has been corrected, thanks.

Reviewer 3: This manuscript reports a sequential methodology for the concentric triple functionalization of graphene patterned surfaces. The controlled one-step chemical patterning of graphene has already been achieved by several groups, including a previous communication by the authors of the current work related with the reported research (ref. 24 of the manuscript). However, this work represents a breakthrough, allowing the step by step covalent modification of 2D patterned graphene by a combined electron-beam lithography/reductive protocol. The sequence of consecutive arylation, deuteration and chlorination reactions was monitored by statistical Raman spectroscopy, and the evaluation of the corresponding bond strengths investigated in thermal detaching experiments. Overall the research is solid and the presentation of results concise and clear with up to date references, only a few minor aspects might deserve further discussion/elaboration:

- We highly appreciate the time and effort that reviewer has invested to review our manuscript and we are very pleased about his/her highly positive evaluation and his/her constructive suggestions are very helpful for an improvement of our manuscript.

Comment 1: “The authors state that a high degree of functionalization is achieved. Could this comment be quantified? Is the surface fully covered by the different functional groups? Differences in the reactivity of diazonium salts vs. D₂O vs. ICl must have been observed.”

Answer: Following the reviewer’s suggestion, the degree of functionalization has been quantified based on our previously introduced method (J. Englert *et al.*, *ACS Nano*, **2013**, *7*, 5472-5482) and the respective information including a brief discussion has been added into supporting information section S4 (including a new Table S2). On the basis of this method, we also quantified the degree of functionalization of previously reported samples on patterning functionalization of graphene for comparison (data provided in table S2). Obviously, our reaction sequence provides a significantly higher degree of functionalization. Nevertheless, according to our quantified results, the surface of the graphene is not fully covered by the functional entities – this is due to steric repulsion of the addends.

The reviewer is right that differences in reactivity of arylation, deuteration, and chlorination were observed. The corresponding differences in reactivity were already systematically investigated in the supporting information S2 (initial version). Specifically, based on our reductive activation concept, the reactivity between the respective diazonium salt and ICl exhibits only a slight difference, whereas D₂O exhibits significantly higher reactivity as shown in Raman spectra (Figure S2, S3, S4).

Comment 2: “In the legend of Figure 1 it is mentioned that “strain-free antaratopic additions provided by surface atoms (O, H) of the underlying SiO₂/Si substrate were enabled”. This aspect, affecting reactivity, should be briefly mentioned in the text.”

Answer: Following the reviewer's suggestion the following description was added into the main text (page 3, see revised manuscript): "*The usage of a reactive Si/SiO₂ substrate – reactivity provided by the presence of accessible surface atoms (etc. O, H) – is a fundamental prerequisite as it allows for a strain-free antaratopic graphene addition scenario, which is in sharp contrast to supratopic addition reactions that would generate enormous strain in the graphene lattice.*^[26-28]. The influence of the Si/SiO₂ has been investigated in the cited preceding studies.

Comment 3: "After e-beam lithography graphene might be a reactive surface, at least for the arylation process. Control experiments before activation with Na/K should be carried out and discussed."

Answer: According to the reviewer's suggestion a reference experiment has been carried out. Here, the patterned graphene (after EBL treatment) with a periodic pattern of holes with 10 μm diameter was directly reacted with 4-bromobenzene diazonium-tetrafluoroborate, D₂O, and ICl under identical conditions (including subsequent work-up) but without prior reductive activation. The corresponding Raman spectra are shown in Figure R2 (below). It can be clearly seen that the Raman D-bands of such samples remained unchanged, suggesting that the reaction did not occur without reductive activation even though EBL may also slightly activate the graphene surface. For the sample reacted with 4-bromobenzene diazonium-tetrafluoroborate, the D band increased slightly with I_D/I_G ratio of 0.13 (before reaction the I_D/I_G < 0.1), indicating a very low degree of functionalization compared with our reductive strategy (I_D/I_G ratio is 2.6). This low degree of functionalization is consistent with previous results (R. Sharma *et al.*, *Nano Lett.* **2010**, *10*, 398-405; G. L. C. Paulus *et al.*, *Acc. Chem. Res.* **2013**, *46*, 160-170; M. A. Bissett *et al.*, *RSC Adv.*, **2014**, *4*, 52215-52218) and is based on the high electron-acceptor character of the diazonium group. The reference experiments and the related discussions were added into the supporting information section S5.

Figure R2. Optical images of patterned graphene with periodic dots of 10 μm diameter (A, B, C). The blue cyclic dots are areas where PMMA (purple) has been removed by EBL such that the graphene become exposed. The Raman spectra of the EBL-exposed graphene zones before and after reaction (without initial reductive activation) with 4-bromobenzene diazonium-tetrafluoroborate (D), D_2O (E), and ICl (F). Scale bar: 5 μm .

Comment 4: “In the cleaning process between subsequent functionalization processes benzonitrile is added to “remove the remaining negative charge on graphene”. Does not remain attached to the surface?”

Answer: The patterned PMMA mask is removed after each covalent functionalization sequence and a fresh PMMA double-layer is installed prior the next EBL-patterning cycle – this is due to the fact that the PMMA layer becomes partly damaged by the addition of the electron-trapping reagent benzonitrile.

We thank the referee for bringing our attention to this highly important point and we are sorry that we didn’t outline the novel patterning/functionalization procedure in every single detail. This has now been corrected and we added substantial information to the main manuscript and to the electronic supplementary information to clarify each process step.

In detail, after quenching the residual negative charges with benzonitrile, the samples were rinsed with 20 mL ethanol and the PMMA layer was removed by acetone vapor (60 min). This leads to a complete removal of benzonitrile.

Reviewer #2 (Remarks to the Author):

Wei et al. have provided a response to the Reviewers' comments to their manuscript titled "Molecular Embroidery of Graphene". In the reviews, three points were consistently made by the Reviewers: (1) Reviewers 1 & 2 stated that the work was not novel considering all the chemistry had already been shown. (2) Reviewers 1&2 questioned the value of the pattern of concentric circles with relatively large feature dimensions. And (3) Reviewers 2 & 3 requested that the degree of functionalization be quantified. Despite the rather lengthy response, none of these issues were adequately addressed and remain outstanding and unresolved in the resubmitted manuscript.

1. Novelty of the chemistry. The authors acknowledge in the first comment to Reviewer 1 that the chemistry is known. The authors make several points in response. (i) "Multifunctionalization of graphene localized in predefined areas remains an unresolved" challenge. This does not address the lack of novelty of the chemistry, rather they are arguing that the lack of novelty in the chemistry can be ignored because the pattern is what is useful (this point will be addressed below). This response is non-responsive to the Reviewer's point. (ii) "One obvious shortcoming is the moderately low degree of functionalization in the reported approaches and which can be overcome by our recently established activation concept of substrate deposited monolayer graphene." However, this still does not address the problem that all the chemistry has been reported previously. (iii) They showed addition of deuterium to the surface, which has not been shown before using this chemistry. It is unclear why deuterium is an interesting addend, particularly when the authors state, in the same response "an even more important drawback is that the addends currently available for patterning graphene are very limited (restricted to hydrogen, aryl, and cis-diene entities." It does not seem that adding deuterium to the library of possible addends substantially expands the scope of materials that can be immobilized onto graphene (although it should be noted that graphene has been functionalized with deuterium previously J. Vac. Sci. Tech. B, 2019, 37, 041804). As such, the chemistry is still not novel, and in addressing this point, the authors continually fail to address the actual points being raised, namely that the chemistry has all been reported before. Reviewer 2 went into great detail to spell out exactly how the chemistry did not break any new ground, and rather than address these points, the authors state that they "are thankful for this reviewer's very positive feedback", while completely failing to address the chemistry issue (see text above "Comment 1" for Reviewer 2).

2. Pattern is neither useful nor groundbreaking. There are three major issues with the multicomponent pattern shown that makes this work unsuitable for Nat. Commun. (i) The difficulty with which the pattern is made. The pattern involves the successive use of photomasks, washing, and addition of new PMMA layers for each new addend. This is a very complex and time-consuming method, and is unlikely to be more widely adopted. (ii) The dimensions of the actual pattern. The authors acknowledge that the "reposition accuracy is only on the micrometer scale". This is a fundamental problem with this patterning approach because of realignment fidelity as the substrate is removed, washed, and realigned for each new addend. The authors state "if special care is taken on the reposition accuracy, one would rather design and align smaller write fields for which we would expect indeed accuracies below 100 nm in x- and y-coordinates." This is simply not true, in particular if the substrate must be continuously removed from the lithography tool. Are the authors suggesting that they did not take special care in this work? For a journal like Nat. Commun. the authors cannot simply allude to possible metrics that can be achieved, they must demonstrate them. And here, the pattern metrics are not great and have likely reached a fundamental limit. The authors acknowledge later that there are fundamental limits to going to narrower spatial resolution, such as "it is not obvious that chemical reagents can reach down to the surface in such narrow resist pits" and that the "PMMA resist sufficiently sticks to the graphene surface", and "going to narrower patterns will be a fundamental technological challenge". Thus they acknowledge huge barriers to using this technique so that patterns with interesting dimensions can be made, which directly contradicts the statement that making smaller patterns "represents in principle fine-tuning of our general patterning/functionalization protocol which goes beyond the scope of this manuscript." This Reviewer strongly disagrees on both points. The authors do not seem to recognize the major challenges that would be required to shrink the dimensions, and unless they show it, then the major limitations should be acknowledged as such in this manuscript. (iii) The utility of the pattern. The authors describe three concentric rings of

functionalized graphene, where in each ring there is a different addend immobilized. It is hard to imagine any application where this will be useful. Reviewer 2 specifically asked the authors to name just such an application. The authors state "We are convinced that our discovery has great potential in opto-electronics, molecular recognition as well as for magnetic-, catalytic- and sensor applications, as we have already stated in the manuscript." However, the authors being convinced is not the standard for publication. It remains entirely unclear how concentric rings of these addends or any others will lead to any useful or groundbreaking application. Again, for a journal like Nat. Commun., an incremental advance is not suitable, and the expectation would be for the demonstration rather than an allusion to the actual novelty.

3. Immobilization density. The authors repeatedly pointed to their excellent grafting density as a justification for publishing this work. In the response, the authors state "One obvious shortcoming [of the work of others] is the moderately low degree of functionalization inherent in the reported approaches and which now can be overcome by our recently established activation concept of substrate deposited monolayer graphene". Later they state "Nevertheless, considering the advantage of our direct reductive activation concept, providing a very high degree of functionalization, zone Ib, has been very highly functionalized – probably up to saturation" and "the reductive activation concept enables a high degree of deuteration of graphene which we have demonstrated with bulk graphene chemistry" and "this allowed us to obtain high degrees of functionalization" and they actually added the comment to the manuscript "the achieved degrees of functionalization are comparable to that of deuterated bulk graphite and significantly exceed the numbers reported for deuterated multi-layer graphene obtained by alternative functionalization approaches."

In section "S4. Raman characterization of functionalized areas: Reduction time and degree of functionalization", the authors report degrees of functionalization based upon Raman spectroscopy, but it seems that only zone II has a degree of functionalization that is significantly higher than previous studies by other groups, with zone II being deuterium, which is by far the smallest of all addends, and far smaller than those to which it is being compared. Is this a fair comparison? It should be noted that while the reviewers critique the degree of functionalization of the previously reported deuterium paper (J. Vac. Sci. Tech. B), the metrics from this paper do not appear in Table S2. Finally, regarding deuterium, their degree of functionalization only is ~1%. How can they claim that this is "probably up to saturation"? Further, for the larger addends, the degree of functionalization seems low and comparable to other methods.

Reviewer #3 (Remarks to the Author):

I'm glad to inform that the authors have fully addressed the comments raised by this reviewer in the initial examination of their article. They also respond in this revised version to the additional queries raised by other reviewers. The paper now presents clear and detailed experimental evidences of the novel multiply patterned-graphene method developed by the authors, and express in a realistic perspective the state of the art in the field and the possible extension of the work to other 2D-patterns. On my view the paper is now ready to be published.

Point-by-Point to Reviewer 2

Reviewer #2: Wei *et al.* have provided a response to the Reviewers' comments to their manuscript titled "Molecular Embroidery of Graphene". In the reviews, three points were consistently made by the Reviewers: (1) Reviewers 1 & 2 stated that the work was not novel considering all the chemistry had already been shown. (2) Reviewers 1&2 questioned the value of the pattern of concentric circles with relatively large feature dimensions. And (3) Reviewers 2 &3 requested that the degree of functionalization be quantified. Despite the rather lengthy response, none of these issues were adequately addressed and remain outstanding and unresolved in the resubmitted manuscript.

1. Novelty of the chemistry. The authors acknowledge in the first comment to Reviewer 1 that the chemistry is known. The authors make several points in response. (i) "Multifunctionalization of graphene localized in predefined areas remains an unresolved" challenge. This does not address the lack of novelty of the chemistry, rather they are arguing that the lack of novelty in the chemistry can be ignored because the pattern is what is useful (this point will be addressed below). This response is non-responsive to the Reviewer's point. ii) "One obvious shortcoming is the moderately low degree of functionalization in the reported approaches and which can be overcome by our recently established activation concept of substrate deposited monolayer graphene." However, this still does not address the problem that all the chemistry has been reported previously.

Reply: In our opinion, the term "novelty" cannot be reduced to the fact that a novel, up to that point never described chemical reaction is the only object of consideration that justifies a publication in a scientific journal. Every scientific progress in chemistry is based on a set of concepts and known chemical reactions that are used in a creative and synergistic fashion to push the boundaries of the state of the art a little bit further.

With respect to our present work: We report on a proof of concept finding on a pioneering and versatile reaction protocol for a geometrically controlled, tailor made multi-functionalization of graphene. Up to now, not even the possibility existed to realize such a covalent attachment of chemical entities on graphene - in distinct areas and distinct shapes - with a complementary profile of properties of the different substituents. Only the creative combination of a suitable chemical reaction scheme with appropriate physical concepts has opened to the door to this new type of 2D-architecture and from our point of view this fact deserves to be described by the term "novelty". Generally, it has been accepted that any newly discovered covalent reaction scheme that can be used for graphene patterning has to be regarded as a breakthrough that justifies the publication in high ranking scientific journals (*Balog, R. et al. Nature Mater. 2010, 9, 315-319; Z. Sun et al., Nat. Commun. 2011, 2, 559-564; F. M. Koehler et al., Angew. Chem. Int. Ed. 2009, 48, 224-227;*).

In addition, for the initial proof of concept for the patterning protocol and to

implement an EDS marker we referred to a well investigated electrophilic trapping reagent, used in bulk reductive graphene activation/functionalization – namely aryl diazonium compounds (an alternative choice would be aryl- or alkyl halides). Beyond that, we extended the scope of potential electrophilic trapping reagents to iodine monochloride to covalently graft chlorine atoms on graphene in a very mild and controllable fashion – a reaction that has not been reported earlier, and a substituent that could in principle easily be exchanged in subsequent substitution reactions with other functional entities.

So what is the breakthrough that we have accomplished with the present work outlined in this manuscript? We have developed for the very first time a multiply-patterned-graphene architecture. Our 2D-hetero-architecture engineering methods cover two aspects: a) it not only contains different functionalities, among which the chlorination with iodine monochloride reported here represents a completely new functionalization schemes and b) it opens the door to access covalently functionalized single-layer graphene, where addends based on different classes of compounds can be directed into restricted, specific areas of any geometrical shape in a repeatable pattern.

Reviewer #2: (iii) They showed addition of deuterium to the surface, which has not been shown before using this chemistry. It is unclear why deuterium is an interesting addend, particularly when the authors state, in the same response “an even more important drawback is that the addends currently available for patterning graphene are very limited (restricted to hydrogen, aryl, and cis-diene entities.” It does not seem that adding deuterium to the library of possible addends substantially expands the scope of materials that can be immobilized onto graphene (although it should be noted that graphene has been functionalized with deuterium previously J. Vac. Sci. Tech. B, 2019, 37, 041804). As such, the chemistry is still not novel, and in addressing this point, the authors continually fail to address the actual points being raised, namely that the chemistry has all been reported before. Reviewer 2 went into great detail to spell out exactly how the chemistry did not break any new ground, and rather than address these points, the authors state that they “are thankful for this reviewer’s very positive feedback”, while completely failing to address the chemistry issue (see text above “Comment 1” for Reviewer 2).

Reply: We have chosen in our patterning example the corresponding substituents on the basis of the following considerations. I) The introduced addend can be identified and characterized by the present set of analytical tools. II) The patterning/activation/functionalization protocol should be applicable for a wide span of different classes of compounds. III) The physical properties of the covalently grafted addends should be complementary (electron donating, electron withdrawing, neutral). IV) The introduction of addends which could subsequently be exchanged by subsequent chemical transformations can enhance the spectrum of potential applications.

With phenyl substituents, chlorine atoms and deuterium atoms all these points can nicely be cover and that may justify why we have judged deuterium as an interesting

addend and have implement it into our patterned architecture

Reviewer #2: 2. Pattern is neither useful nor groundbreaking. There are three major issues with the multicomponent pattern shown that makes this work unsuitable for Nat. Commun. (i) The difficulty with which the pattern is made. The pattern involves the successive use of photomasks, washing, and addition of new PMMA layers for each new addend. This is a very complex and time-consuming method, and is unlikely to be more widely adopted.

Reply: As described above, up to present no method existed to covalently attach functional entities on graphene restricted to specific areas in an arbitrary geometrical shape. In addition, by this method different physical properties can be “written” directly next to each other or can be separated by an area of un-functionalized graphene. The referee is right, that this proof of concept protocol is not based on a simple one pot reaction but utilizes iterative protection, patterning, functionalization steps. E-beam lithography still stands as the most commonly used approach for covalent patterning of graphene as demonstrated by several preceding examples for one-step based covalent patterning techniques for graphene (F. M. Koehler *et al.*, *Angew. Chem. Int. Ed.* **2009**, *48*, 224-227; Z. Sun *et al.*, *Nat. Commun.* **2011**, *2*, 559-564; J. Li *et al.*, *J. Am. Chem. Soc.* **2016**, *138*, 7448-7451; T. Wei *et al.*, *ChemPlusChem*, **2020**, *85*, 1655-1668)). In this sense, the complexity of a method will not necessarily affect its use - the construction of wafer based electrical circuits is also built on a multistep photo masking process. With our study we only can present a novel protocol to realize up-to-now not achievable 2D-architectures based on chemically functionalized graphene and cannot provide a fully optimized, easy to handle process ready to implement in industry.

Reviewer #2: (ii) The dimensions of the actual pattern. The authors acknowledge that the “reposition accuracy is only on the micrometer scale”. This is a fundamental problem with this patterning approach because of realignment fidelity as the substrate is removed, washed, and realigned for each new addend. The authors state “if special care is taken on the reposition accuracy, one would rather design and align smaller write fields for which we would expect indeed accuracies below 100 nm in x- and y-coordinates.” This is simply not true, in particularly if the substrate must be continuously removed from the lithography tool. Are the authors suggesting that they did not take special care in this work? For a journal like Nat. Commun. the authors cannot simply allude to possible metrics that can be achieved, they must demonstrate them. And here, the pattern metrics are not great and have likely reached a fundamental limit. The authors acknowledge later that there are fundamental limits to going to narrower spatial resolution, such as “it is not obvious that chemical reagents can reach down to the surface in such narrow resist pits” and that the “PMMA resist sufficiently sticks to the graphene surface”, and “going to narrower patterns will be a fundamental technological challenge”. Thus they acknowledge huge barriers to using this technique so that patterns with interesting dimensions can be made, which directly contradicts the statement that making smaller patterns “represents in principle fine-tuning of our general patterning/functionalization protocol which goes beyond the scope of this manuscript.” This Reviewer strongly disagrees on both points. The authors

do not seem to recognize the major challenges that would be required to shrink the dimensions, and unless they show it, then the major limitations should be acknowledged as such in this manuscript.

Reply: For our described patterning setup we have used a very large write-field (2.000 μm x 2.000 μm) to create 40 x 40 circles. We did not focus on the achievable dimensions and didn't want to push the method to any possible limit. When we shift to smaller write-fields with a lesser amount of repetitive geometrical structures we could definitely increase the resolutions and obtain structures in the sub 100 nm domain. This is not only a guessed figure but is justified by the fact that accuracy in the nanometer scale dimension can be achieved with e-beam lithography (Y. Q. Zheng, *et al*, *Adv. Mater. Technol.* 2017, 2, 1600237-1600249. C. Vieu, *et al*, "Electron beam lithography: resolution limits and applications", *Applied Surface Science*, Volume 164, Issue 1, p. 111-117).

We most certainly realize that there are challenges that have to be overcome when the dimensions of the written patterns are shrunk substantially and we have already mentioned that the creeping of reagents under the PMMA resist may become an issue as it has been observed for microscale patterns (A. Hirsch *et al.*, *Angew. Chem. Int. Ed.* 2020, 59, 5602-5606). To what extent this may be a problem and if this or other obstacles can be handled cannot be predicted and has indeed to be elucidated by respective experiments. For our initial proof of concept report this was not the main focus.

To the editor: To explore the possible limitations of our novel patterning/functionalization protocol in detail was not the scope of our initial manuscript. If such an investigation is needed to justify a publication in Nature Communication we are most certainly willing to set up the respective experiments and to provide the specific data.

Reviewer #2: (iii) The utility of the pattern. The authors describe three concentric rings of functionalized graphene, where in each ring there is a different addend immobilized. It is hard to imagine any application where this will be useful. Reviewer 2 specifically asked the authors to name just such an application. The authors state "We are convinced that our discovery has great potential in opto-electronics, molecular recognition as well as for magnetic-, catalytic- and sensor applications, as we have already stated in the manuscript." However, the authors being convinced is not the standard for publication. It remains entirely unclear how concentric rings of these addends or any others will lead to any useful or groundbreaking application. Again, for a journal like Nat. Commun., an incremental advance is not suitable, and the expectation would be for the demonstration rather than an allusion to the actual novelty.

Reply: Applications based on three simple concentric rings of functionalized graphene indeed do not seem very likely. Given the fact, that the geometric shape of the functionalized area is not restricted to round circles and that any arbitrary shape of functionalization can be written – insulating paths based on covalently functionalized sp^3 tracks with neutral substituents, p-n junctions of arbitrary geometrical shape and with varying dimension – our approach may become highly interesting in the field of nanostructured electronics.

a) Field of application – nanostructured electronics: With our set of complementary addends – substituted benzene, chlorine and deuterium atoms – comprising a set of

different electronic properties (electro-withdrawing, electron-donating, neutral) the overall properties of the constructed 2D-hetero-architecture can be tailored. Regular patterns of hydrogen/deuterium-covered regions result in a confinement the charge carriers in the pristine graphene regions, thus leading to a bandgap opening and consequence a n-type semiconductor (*Balog, R. et al. Nature Mater. 2010, 9, 315-319.*). The bandgap depends on the amount of attached hydrogen/deuterium atoms. It has been shown that chlorination leads to a p-doped graphene, which is complementary to the n-doping achieved by deuteration (*X. Zhang, et al. ACS Nano 2013, 7, 7262-7270.*). By the attachment of differently substituted aromatic substituents (σ/π -donor or acceptor) a specific geometrically defined region of the graphene lattice can be addressed directly and fine-tuned with respect to its electrical properties. The properties of these type of chemically written p-n junctions may further be fine-tuned the control of the degree of functionalization in each addend zone. Semiconductor p-n junctions are elementary building blocks of many electronic devices such as transistors, solar cells, photodetectors, and integrated circuits and a potential application of tailor-made graphene p-n junction – based on our patterning/activation/functionalization protocol – seems, for the current point of view reasonable and provides a complementary approach to silicon based wafer technologies.

b) Field of application - opto-electronics: With our patterning/activation/functionalization protocol we have shown that it is possible to write phenyl-based substituents to distinct areas on the graphene flake. As aromatic systems are common components in organic solar cells our structuring/functionalization technique may open the door to 2D-architectures, where light of different wavelengths (based on the type of attached chromophore) can be harvested in different areas on the same electrical conducting substrate. With the implementation of nanostructured electronics this concept can be extended to build light sensitive sensors on a nanometer scale.

c) Field of application - information storage with molecules: In our days, information is stored based on a two-state electrical process. Our innovative approach of writing chemical information in a predefined pattern may become of great importance in a novel field of information storage based on distinct chemical molecules where more than only two states (on/off) of information can be stored. Most certainly, this cannot be realized by the described iterative lithography structuring/chemical activation-functionalization approach but our presented protocol shows that it is possible to covalently craft different chemical entities in a controlled fashion in predefined graphene regions.

To the editor: The mentioned three examples represent only some detailed reflections about possible applications – besides many others. Currently we are in contact with colleges in our engineering department in order to set up experiments in field a) – writing of electronic band-gap and p-n-structures based on our outlined protocol. But in our opinion the demonstration of an unprecedented and successive spatially resolved 2D chemistry on an atomically thin material constitutes a major breakthrough that can stand by itself in Nature Communication without the proof of a real technical application.

Of course, in a revised version of the manuscript we have described the potential of our approach with respect to possible applications in the fields of a) nano-electronics, b) opto-electronics, and c) information storage with chemicals, as outlined above, in more

detail.

Reviewer #2: 3. Immobilization density. The authors repeatedly pointed to their excellent grafting density as a justification for publishing this work. In the response, the authors state “One obvious shortcoming [of the work of others] is the moderately low degree of functionalization inherent in the reported approaches and which now can be overcome by our recently established activation concept of substrate deposited monolayer graphene”. Later they state “Nevertheless, considering the advantage of our direct reductive activation concept, providing a very high degree of functionalization, zone Ib, has been very highly functionalized – probably up to saturation” and “the reductive activation concept enables a high degree of deuteration of graphene which we have demonstrated with bulk graphene chemistry” and “this allowed us to obtain high degrees of functionalization” and they actually added the comment to the manuscript “the achieved degrees of functionalization are comparable to that of deuterated bulk graphite and significantly exceed the numbers reported for deuterated multi-layer graphene obtained by alternative functionalization approaches.”

In section “S4. Raman characterization of functionalized areas: Reduction time and degree of functionalization”, the authors report degrees of functionalization based upon Raman spectroscopy, but it seems that only zone II has a degree of functionalization that is significantly higher than previous studies by other groups, with zone II being deuterium, which is by far the smallest of all addends, and far smaller than those to which it is being compared. Is this a fair comparison? It should be noted that while the reviewers critique the degree of functionalization of the previously reported deuterium paper (J. Vac. Sci. Tech. B), the metrics from this paper do not appear in Table S2. Further, for the larger addends, the degree of functionalization seems low and comparable to other methods.

Reply: The key advantage of a reductive activation of graphene is that it can achieve a high degree of functionalization in the bulk material, including not only graphene, but also other carbon allotropes (carbon nanotubes and fullerene). (*Nature Chemistry* 2011, 3, 279-286; *Acc. Chem. Res.* 2013, 46, 87-96; 2019, 52, 2037-2045; *J. Am. Chem. Soc.* 2011, 133, 19459-19473; 2011, 133, 7985-7995; 2013, 135, 18385-18395; 2016, 138, 15642-15647; 2016, 138, 1647-52; 2017, 139, 5175-5182; 2017, 139, 11760-11765; 2018, 140, 3352-3360; 2020, 142, 2327-2337. *Angew. Chem. Int. Ed.* 2013, 52, 754-757; 2016, 55, 5861-5864; 2017, 56, 12184-12190. 2018, 57, 4338-4354; 2019, 58, 816-820; 2019, 131, 8142-8146; 2020, 132, 5651-5655; *Nat. Commun.* 2016, 7, 2041-1723; 2017, 8, 2041-1723.)

As we have clearly demonstrated this approach can be transferred from the bulk material to a single layer of graphene and provides a method to covalently functionalize mono-layer graphene in highly homogeneous fashion. Besides the actual degree of functionalization, the homogeneity of the addend coverage represents another highly important factor and has already been emphasized in our manuscript (“In addition, the rather uniform distribution of the I_D/I_G ratio within the patterned region clearly speaks for a homogeneous addend binding.” Section of Statistical Raman analysis; page 6. Both initial/revised manuscript). As we do take comments on this critical point very seriously, we have carried out a reference experiment based on a large single-layer graphene area (0.5

× 0.5 cm). The sample was reductively activated and deuterated as outlined in the manuscript. It can clearly be seen that upon a reduction time of 50 min, a very pronounced D-band is obtained (Figure R1), comparable to the spectral information presented for the patterned functionalization in Supplementary Figure 6 after 90 min reduction time. This is a clear indication for the generation of a high amount of sp^3 -carbon centers in the respective graphene area (Lucchese *et al.*, "Quantifying ion-induced defects and Raman relaxation length in graphene", *Carbon* **2010**, *48*, 1592-1597; Cançado, *et al.*, "Quantifying defects in graphene *via* Raman spectroscopy at different excitation energies", *Nano Lett.*, **2011**, *11*, 3190-3196) and can be interpreted as a high degree of deuteration of this mono-layer graphene (Figure R1). Compared to the patterned graphene deuteration, the shorter reduction time (for the unpatterned deuteration scenario) may be explained by the easier reduction of graphene without PMMA covering.

Figure R1. Raman spectra for the reductive deuteration of a large-scale graphene (0.5 × 0.5 cm) upon different reduction times.

As we have stated in the supporting information, we only compared the obtained data with respect to the degree of functionalization of our covalently **patterned** graphene with data obtained for other **patterned** graphene samples. The publication of *A. R. Mazza et al., J. Vac. Sci. Technol. B, 2019, 37, 041804* only deals with the **unpatterned** deuteration of graphene and that is the only reason why it is not listed in Supplementary Table 2 – a direct comparison between an unpatterned and a patterned sample seems not to be fair.

To the editor: In our opinion, as there is no direct correlation between the ID/IG ratio and an absolute value for a degree of functionalization a direct quantitative comparison (based on distinct degree of functionalization values) of our spectral data obtained for a functionalized patterned mono-layer graphene and the spectral data of other groups obtained for bulk functionalized materials or unpatterned graphene could easily lead to a misinterpretation of the results and therefore, we haven't provided a direct comparison of these values in Supplementary Table 2.

In a revised version of the manuscript we have implemented the reference experiment in the supplementary information section S5 and provide a short discussion where we

outlined the challenges of comparing Raman spectroscopic data obtained for bulk functionalized materials with the respective data of functionalized patterned mono-layer graphene and unpatterned graphene.

Reviewer #2: Finally, regarding deuterium, their degree of functionalization only is ~1%. How can they claim that this is “probably up to saturation”?

Reply: We are sorry that we have used the term “saturation” in this context as it could easily be misinterpreted. With “saturation” we wanted to express that the reaction itself has reached a maximum and not that an achievable quantitative degree of functionalization (upper limit) has been obtained – unfortunately, we haven’t made this clear.

Upon the reductive activation strategy, the functionalized graphene indeed provides a relatively high degree of functionalization, which is based on an antaratopic addition mode utilizing a reactive Si/SiO₂ substrate. However, when it comes to extremely high degrees of functionalization (e.g. deuteration), the underlying Si/SiO₂ substrate cannot provide sufficient reaction centers to release the accumulated strain energy provide by the supratopic attachment of the addends, in full extend. Thus, the reaction reaches its limit and this is what we denoted with the term “saturation”.

To the editor: In a revised version of the manuscript, we avoided the using of the term “saturation” in order to circumvent any possible confusion.

Reviewer #3 (Remarks to the Author): I’m glad to inform that the authors have fully addressed the comments raised by this reviewer in the initial examination of their article. They also respond in this revised version to the additional queries raised by other reviewers. The paper now presents clear and detailed experimental evidences of the novel multiply patterned-graphene method developed by the authors, and express in a realistic perspective the state of the art in the field and the possible extension of the work to other 2D-patterns. On my view the paper is now ready to be published.

Reply: We are very grateful to this reviewer for agreeing to the publication of our manuscript and also the time and effort spent in reviewing our manuscript.